# Immune-suppression by OsHV-1 viral infection causes fatal bacteraemia in Pacific oysters

Julien de Lorgeril[1], Aude Lucasson[1], Bruno Petton[2], Eve Toulza[1], Caroline Montagnani[1], Camille Clerissi[1], Jeremie Vidal-Dupiol[1], Cristian Chaparro[1], Richard Galinier[1], Jean-Michel Escoubas[1], Philippe Haffner[1], Lionel Dégremont[3], Guillaume M. Charrière[1], Maxime Lafont[1], Abigaïl Delort[1], Agnès Vergnes[1], Marlène Chiarello [4], Nicole Faury[3], Tristan Rubio[1], Marc A. Leroy[1], Adeline Pérignon[5], Denis Régler[5], Benjamin Morga[3], Marianne Alunno-Bruscia[2], Pierre Boudry[6], Frédérique Le Roux[7], Delphine Destoumieux-Garzón[1], Yannick Gueguen [1] & Guillaume Mitta[1]

Infectious diseases are mostly explored using reductionist approaches despite repeated evidence showing them to be strongly influenced by numerous interacting host and environmental factors. Many diseases with a complex aetiology therefore remain misunderstood. By developing a holistic approach to tackle the complexity of interactions, we decipher the complex intra-host interactions underlying Pacific oyster mortality syndrome affecting juveniles of *Crassostrea gigas*, the main oyster species exploited worldwide. Using experimental infections reproducing the natural route of infection and combining thorough molecular analyses of oyster families with contrasted susceptibilities, we demonstrate that the disease is caused by multiple infection with an initial and necessary step of infection of oyster haemocytes by the *Ostreid herpesvirus* OsHV-1 µVar. Viral replication leads to the host entering an immune-compromised state, evolving towards subsequent bacteraemia by opportunistic bacteria. We propose the application of our integrative approach to decipher other multifactorial diseases that affect non-model species worldwide.

[1] IHPE, Université de Montpellier, CNRS, Ifremer, Université de Perpignan Via Domitia, Place E. Bataillon, 34095 Montpellier, France. [2] LEMAR UMR 6539, UBO/CNRS/IRD/Ifremer, 11 presqu'île du vivier, 29840 Argenton-en-Landunvez, France. [3] Laboratoire de Génétique et Pathologie des Mollusques Marins, Ifremer, Avenue du Mus de Loup, 17930 La Tremblade, France. [4] Marine Biodiversity, Exploitation and Conservation (MARBEC), Université de Montpellier, CNRS, IRD, Ifremer, Place E. Bataillon, 34095 Montpellier, France. [5] CRCM, Comité de la Conchyliculture de Méditerranée, Quai Baptiste Guitard, 34140 Mèze, France. [6] LEMAR UMR6539, CNRS/UBO/IRD/Ifremer, ZI pointe du diable, CS 10070, F-29280 Plouzané, France. [7] Sorbonne Universités, UPMC Paris 06, CNRS, UMR 8227, LBI2M, Ifremer, Station Biologique de Roscoff, CS 90074, F-29680 Roscoff, France. These authors contributed equally: Julien de Lorgeril and Aude Lucasson. Correspondence and requests for materials should be addressed to Y.G. (email: ygueguen@ifremer.fr) or to G.M. (email: mitta@univ-perp.fr)

For decades, methodological limitations have restricted the study of infectious diseases to simplified experimental pathosystems in which the influences of host and pathogen diversity and biotic and abiotic environments have been minimized. Such reductionist approaches have made diseases with complex aetiologies difficult to characterize. Thus, there is an incomplete understanding of diseases in which a conserved consortium of micro-organisms co-operates to induce pathogenesis, diseases involving pathogens that cause immune deficiency followed by secondary infections, and diseases that are influenced by a series of host and environmental factors. There is a lack of understanding of some diseases triggering recurrent mass mortalities in non-model species of ecological and/or economic interest, such as pollinators, corals and marine molluscs[1–3]. These dramatic epizooties remain incompletely characterized because epidemiological descriptions require holistic approaches to decipher the whole pathosystem.

The objective of the present work was to examine one disease of complex aetiology affecting one of the most utilized invertebrate species in the world, the Pacific oyster *Crassostrea gigas*, applying a holistic approach to decipher the phenomenon. This species has been introduced from Asia to numerous countries and is now the main farmed oyster species worldwide[4]. Introduced to France in the 1970s, *C. gigas* suffers mass mortalities associated with complex interactions between the host, the environment and pathogens[5]. The severity of these mortality outbreaks has dramatically increased since 2008. They mainly affect juvenile stages, decimating up to 100% of young oysters in French farms[6]. In recent years, this mortality syndrome, called Pacific oyster mortality syndrome (POMS)[7], has become panzootic, being observed in all coastal regions of France and numerous other countries worldwide[6,8].

Research efforts have revealed a series of factors contributing to the disease, including infectious agents interacting with seawater temperature and oyster genetics[6,9–14]. The dramatic increase in mortality since 2008 coincided with the recurrent detection of *Ostreid herpesvirus* (OsHV-1) variants in moribund oysters both in France[14–16] and worldwide[6,17–20]. This increase has naturally driven research efforts to focus on the viral aetiology of the disease. However, the involvement of other aetiological agents is suspected. In particular, strains assigned to the genus *Vibrio* have been shown to be associated with the disease[21]. Among these, a *Vibrio crassostreae* population carrying a plasmid required for virulence has been repeatedly identified in diseased oysters[22]. Recent studies have also proposed that the stability of the natural bacterial microbiota, which is abundantly present in healthy oysters, influences their resistance to stress, or invasion by pathogens[23,24]. However, the roles of host genetics, pathogens, and opportunistic and commensal microbes have been studied with a focus on a restricted number of factors tested in isolation[10,21–27]. Thus, the dynamics, relative weight and interactions of these different parameters in the disease remain to be established.

In the present study, we concomitantly characterize the transcriptional responses of oysters and the dynamics of their associated micro-organisms after exposure to an infectious environment using an experimental infection system that reproduced the natural route of infection in biparental families of oysters selected to display contrasted phenotypes (susceptible vs. resistant). This holistic approach allows us to establish that OsHV-1 replication in haemocytes is the initial step of the infectious process leading to an immune-compromised state of the host, which evolves towards subsequent bacteraemia by opportunistic bacteria. By identifying critical intra-host interactions between micro-organisms and host immunity, this study cracks the code of POMS.

## Results

**Production of oyster families with contrasted phenotypes.** To characterize the complex dynamics determining the outcome of Pacific oyster mortality syndrome, we produced 15 biparental oyster families with highly contrasted resistance phenotypes with regard to the disease. These families were produced from genitors that had experienced different selective filters. The genitors having experienced a high level of natural selection by the disease were either wild oysters collected from farming areas or oysters issued from mass selection programmes[28]. The genitors that had experienced low-level natural selection were wild oysters recruited outside farming areas (Supplementary Table 1). The juvenile offspring of the 15 biparental families produced were subjected to an 'natural' experimental infection mimicking disease transmission in the wild[12,13,29] (Supplementary Fig. 1). High variability in the dynamics of mortalities and percentages of survival (ranging from 0 to 97.4% at 330 h post-infection) were observed among families (Fig. 1). Two families showing highly contrasted phenotypes, Susceptible Family 11 ($S_{F11}$) and Resistant Family 21 ($R_{F21}$) (Mantel–Cox log-rank test, $p < 0.0001$), were selected for the first set of molecular and histological analysis. During the experimental infection, $S_{F11}$ oysters mortality began at 66 h and increased dramatically. At the end of the experiment (330 h), only 0.7% of $S_{F11}$ oysters had survived. In contrast, almost all oysters of the $R_{F21}$ family (97.4%) survived following exposure to the same infectious environment. The survival rates of $S_{F11}$ and $R_{F21}$ oysters were also measured concomitantly in a batch of oysters left on oyster farms; they showed identical phenotypes, with 2 and 98.1% survival after 384 h of exposure to the infectious environment for $S_{F11}$ and $R_{F21}$, respectively. These results confirm that the disease that developed in our experimental set-up resembles the disease contracted in the natural environment with the same outcomes. Thereafter, the dynamics of the disease in $S_{F11}$ and $R_{F21}$ oysters were investigated by thorough molecular analyses.

**OsHV-1 µVar infection occurs early in disease development.** OsHV-1 load and transcriptional activity were monitored in $S_{F11}$ and $R_{F21}$ oysters throughout the experiment using qPCR and RNA-seq, respectively (Fig. 2a). Whereas OsHV-1 DNA and RNA were detected in both families as early as 12 h after the beginning of experimental infection, very intense replication occurred in $S_{F11}$ oysters, with DNA and RNA levels 3-log higher

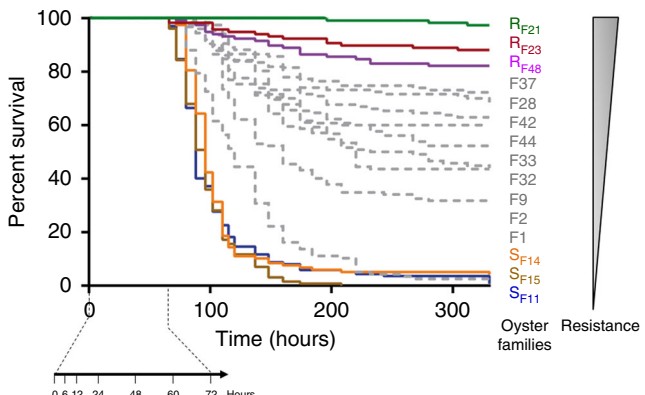

**Fig. 1** Production of oyster biparental families with contrasted resistance phenotypes. Kaplan–Meier survival curves of the 15 families of recipient oysters ($n = 1000$ for each family) during the 'natural' experimental infection. At each time indicated on the arrow, 3 triplicates of 10 oysters were sampled from each tank for further molecular analysis

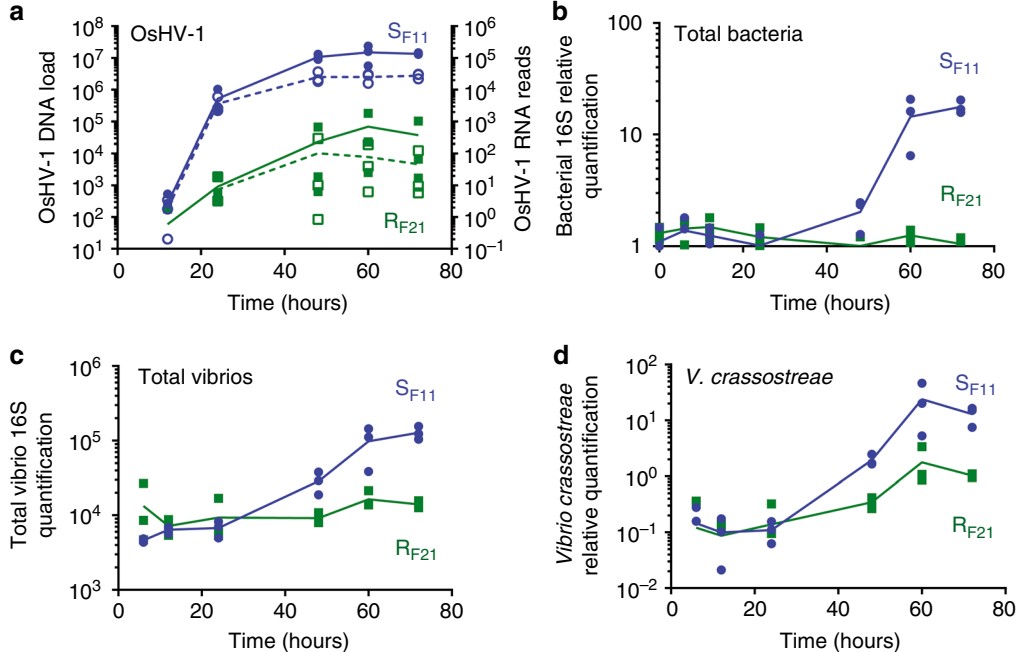

**Fig. 2** OsHV-1 and bacterial colonization in the susceptible $S_{F11}$ oysters during the 'natural' experimental infection. **a** The OsHV-1 load was quantified by qPCR and expressed as viral genomic units per ng of oyster DNA (plain lines, filled symbols); viral replication was estimated by the total number of RNA-seq reads mapped on the OsHV-1 genome (dotted lines, open symbols). Relative quantification of total bacteria (**b**), total *Vibrio* load (**c**) and *Vibrio crassostreae* (**d**) abundance were measured by qPCR. Dots represent distinct pools of 10 oysters. The mean values (plain and dotted lines) are displayed

than in $R_{F21}$ oysters at 24 h after exposure initiation (Fig. 2a). The time-course of OsHV-1 ORF expression is shown in Supplementary Fig. 2 for both oyster families. In $S_{F11}$ oysters, the maximum viral load and transcriptional activity were reached at 48 h and remained stable until the first deaths (at 66 h). The viral load remained low in $R_{F21}$ throughout the experiment without any remarkable mortality until the end of the experiment (330 h). Alignment of the Illumina reads to the OsHV-1 genome available in the NCBI database[30] revealed that the virus used in our experiments corresponded to an OsHV-1 µVar variant, as indicated by (i) the deletion of 3 ORFs (ORF36, ORF37 and part of ORF38), (ii) the deletion of an adenosine upstream of ORF43 and (iii) a 12-nt deletion in the microsatellite locus H10, which are shared characteristics of µVar variants[14,20,31]. These variants have been associated with mass mortalities of juvenile oysters since 2008[6,14,20,32]. Taken together, these results show that OsHV-1 µVar infection occurs in both families, but only $R_{F21}$ oysters successfully control viral replication.

**Dysbiosis and bacteraemia occur in susceptible oysters**. To investigate the dynamics of oyster microbiota in the two families showing contrasting resistance to the disease, we analysed total bacterial communities using 16S rDNA metabarcoding over the first three days of the experiment. Overall, 6,061,881 clusters were obtained from 42 libraries (2 families, 7 sampling times, in triplicate). After cleaning steps and singleton filtering, 4,238,989 sequences affiliated with 10,080 OTUs were kept for further analyses (Supplementary Data 1). A sufficient sequencing depth was confirmed by species richness rarefaction curves (Supplementary Fig. 3). Changes in microbiota composition (Supplementary Fig. 4 and Supplementary Fig. 5) were much higher in $S_{F11}$ oysters than in $R_{F21}$ oysters throughout the experiment. Indeed, 105 OTUs significantly differed in terms of relative proportions in $S_{F11}$ oysters, as opposed to 0 in $R_{F21}$

oysters (Supplementary Fig. 5). A principal coordinate analysis (PCoA) of the Bray–Curtis dissimilarity matrix (beta diversity) consistently revealed a higher microbiota dispersion in the $S_{F11}$ family (multivariate homogeneity of group dispersions, d.f. = 1; $p$ = 0.016) than in the $R_{F21}$ family (Fig. 3a). This disruption of the bacterial community structure occurred in $S_{F11}$ oysters between 24 h and 48 h concomitantly with the active replication of OsHV-1 µVar. Similarly, the Chao1 and Shannon's H indexes of alpha diversity increased significantly in the $S_{F11}$ family during the infectious process (Chao1: analysis of variance (ANOVA), d.f. = 6; $p$ = 1.39e-07 and Shannon's H index: ANOVA, d.f. = 6; $p$ = 3.54e-05), whereas they remained stable in the $R_{F21}$ family (Supplementary Fig. 6). All the bacterial genera that changed significantly over the 'natural' experimental infection in $S_{F11}$ and $R_{F21}$ oysters are reported in Supplementary Data 2. Among these modified genera, those representing more than 4% of the bacteria in at least one sample of $S_{F11}$ oysters are shown in Fig. 3b. The corresponding OTUs, which represent only 2.07% of the total bacteria at the beginning of the experiment, represent 59.07% of the $S_{F11}$ bacterial community at 72 h, when the first mortalities occurred. Some of these OTUs were assigned to the genera *Vibrio* and *Arcobacter*, which have been previously associated with oyster mortalities[22,24]. In contrast, in $R_{F21}$ oysters, the same taxa did not vary significantly over time (Supplementary Fig. 7 and Supplementary Data 2).

In addition, a significant increase in total bacterial abundance was observed in $S_{F11}$ oysters only, which started at 48 h and continued to rise until the end of the experiment (ANOVA, d.f. = 6; $p < 0.0001$; Fig. 2b). Compared with T0, the total bacteria were 13- and 17-fold higher at 60 h and 72 h, respectively, suggesting a massive bacterial proliferation in $S_{F11}$ oysters. Simultaneously, we observed a high increase in total *Vibrio* load and abundance of *Vibrio crassostreae*, a previously characterized bacterial pathogen of oysters associated with the disease[21,22] (Fig. 2c, d). Accordingly, histological analyses revealed bacterial

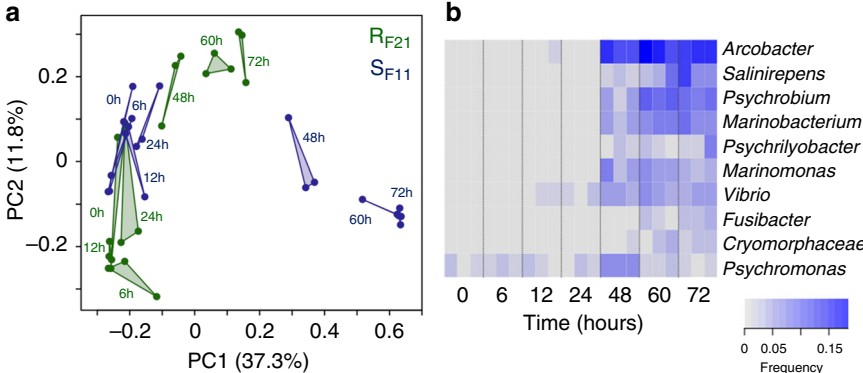

**Fig. 3** Microbiota changes in the susceptible S$_{F11}$ and the resistant R$_{F21}$ oysters during the 'natural' experimental infection. **a** Principal coordinate analysis (PCoA) plot of the Bray–Curtis dissimilarity matrix of the microbiota. Each point of the triangles corresponds to one of the 3 replicates at one kinetic time. Dots represent distinct pools of 10 oysters. **b** Heatmap of the bacterial taxa that were significantly modified in susceptible oysters during the course of infection. Analyses were performed at the genus level. Only genera with a relative proportion superior to 4% in at least one sample are shown. The intensity level of the blue represents the relative abundance of bacterial taxa. At each time, the analysis was performed on 3 distinct pools of 10 oysters

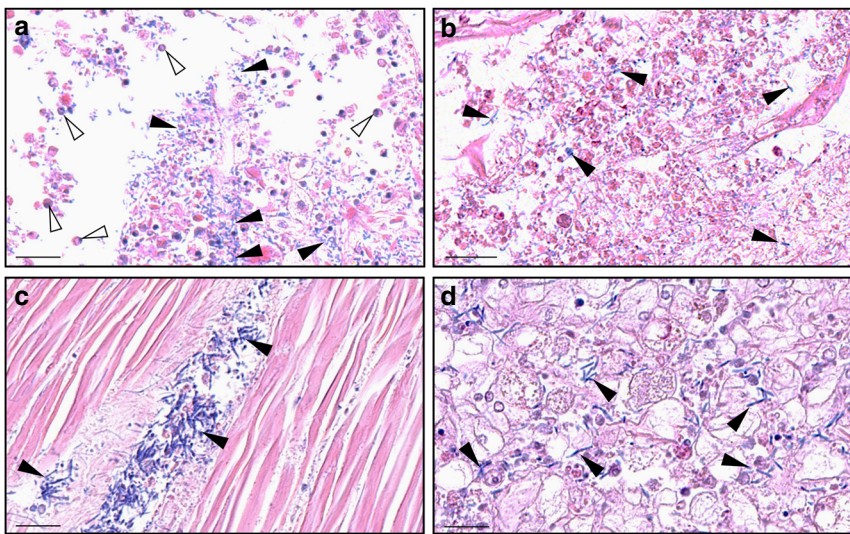

**Fig. 4** Accumulation of bacteria in tissues of the susceptible S$_{F11}$ oysters. Giemsa staining was performed on paraffin wax-embedded tissue histological sections of S$_{F11}$ oysters sampled at different time points to visualize tissue colonization by bacteria. Oyster tissues and cells were coloured in shades of pink to purple, and most bacteria appeared in deep blue (scale bars corresponds to 20 μM). **a** At 54 h after the beginning of the 'natural' experimental infection, bacteria accumulated in the gills of the S$_{F11}$ oysters (filled arrowheads). Rounded cells reminiscent of haemocytes were found in both gill tissues and outside tissues in the vicinity of the bacteria (open arrowheads). At 78 h, gill tissues appeared massively degraded, and bacteria were found in most tissues of the S$_{F11}$ oysters, e.g., in **b** gills, **c** adductor muscle and **d** interstitial tissues near the digestive tract. No bacteria or tissue damage were observed on sections of S$_{F11}$ or R$_{F21}$ either at the beginning of the experimental infection or at any time points for the R$_{F21}$ oyster sections (Supplementary Fig. 8)

colonization and tissue damage in S$_{F11}$ oyster histological sections (Fig. 4). At 54 h, bacterial accumulation accompanied by infiltrating haemocytes was visible in different areas both inside and outside the gill tissues (Fig. 4a). At 78 h, the gill tissue structure was completely disrupted and bacteria had spread throughout the body (Fig. 4b, c, d and Supplementary Fig. 8). In contrast, in R$_{F21}$ oysters, total bacteria and vibrios remained low and stable (Fig. 2b–d), and no tissue damage or tissue colonization by bacteria were observed at any sampled time point (Supplementary Fig. 8).

**Resistant oysters display an early antiviral response**. To identify key host factors controlling the infection, we compared the transcriptomic responses of S$_{F11}$ and R$_{F21}$ oysters by RNA-seq over the time-course of the 'natural' experimental infection. In total, the sequencing of 42 samples (2 families, 7 sampling times, in triplicate) yielded between 20.4 and 32.3 million Illumina paired reads per sample; 69.7–75.3% of them mapped to the *C. gigas* V9 reference genome. RNA-seq results were validated by RT-qPCR on 30 genes at all sampling times in both oyster families ($r^2 = 0.936$) (Supplementary Fig. 9). The transcriptomic responses of S$_{F11}$ and R$_{F21}$ oysters can be divided into two phases: an early response (before 24 h) and a later response (24 h to 72 h) (Fig. 5a). In the early response, the total number of differentially expressed genes was higher in R$_{F21}$ than in S$_{F11}$ oysters (Fig. 5a). To gain access to the functions enriched during this early phase, we used gene ontology (GO) enrichment analyses (Fig. 5b and Supplementary Fig. 10). At these time points, transcriptomes of R$_{F21}$ oysters were more enriched in functional categories related to immunity than S$_{F11}$ transcriptomes (Fig. 5b and Supplementary Fig. 10). Among those genes, 40.9% were involved in antiviral defence (Supplementary

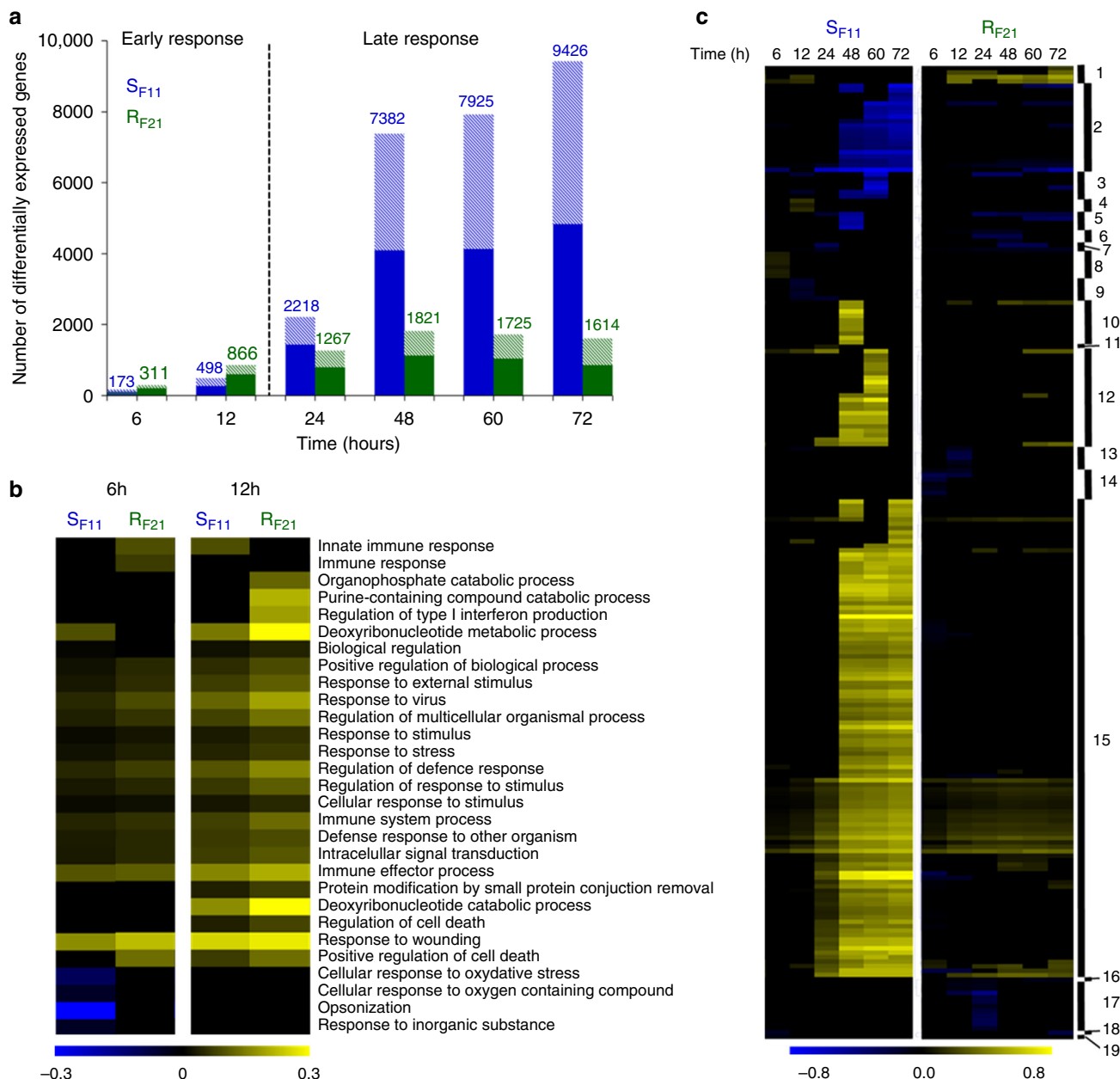

**Fig. 5** The resistant $R_{F21}$ oysters display an early antiviral response that is delayed in the $S_{F11}$ susceptible oysters. **a** The number of genes that were differentially expressed during the 'natural' experimental infection was higher in $R_{F21}$ (green) than in $S_{F11}$ (blue) oysters before 24 h (early response). This trend was reversed after 24 h and until the end of the infection (late response). Upregulated genes are represented as filled coloured bars and downregulated genes are represented as hatched bars. **b** Heatmap focusing on the two clusters containing 29 immune-related GO categories that were significantly enriched at 6 and 12 h (FDR < 1%; clusters A and B, Supplementary Fig. 10). **c** Heatmap of the 220 significantly enriched GO categories (FDR < 1%, biological processes root) clustered according to the Pearson correlation (numbered filled bar). The enrichment intensity was expressed as the ratio between the number of genes that were significantly up- (yellow heat) or downregulated (blue heat) in the category compared with the total number of genes in the category. If the intensity was equal to zero (black heat), then the enrichment was not significant for the corresponding condition. Detailed results (cluster number, GO terms and enrichment values) are presented in Supplementary Data 4

Data 3), encoding elements of the RLR/STING (e.g., RIG-1, IRF and cGAS) and JAK/STAT (e.g., STAT and SOCS2) pathways, antiviral effectors (e.g., IFI44, Viperin and SAMHD-1) and proteins involved in the apoptosis (e.g., TNF and caspase-3) and maintenance of cellular homoeostasis under viral infection (e.g., poly(ADP-ribose) polymerase). Taken together, these data show that the antiviral response is more intense in $R_{F21}$ oysters than in $S_{F11}$ oysters during the first 12 h following exposure to the examined infectious environment.

**The antiviral response of susceptible oysters is inefficient.** During the second phase of the response (after 24 h), $S_{F11}$ oysters displayed a major reprogramming of their transcriptome, with significant changes in the expression of 9426 genes (33.6% of the transcriptome, Fig. 5a) and a large number of enriched functional categories as determined by GO enrichment analyses (Fig. 5c and Supplementary Data 4). This major transcriptomic reprogramming was not observed in $R_{F21}$ oysters (Fig. 5a, c). During this second phase, immunity and antiviral defence genes (cluster 15

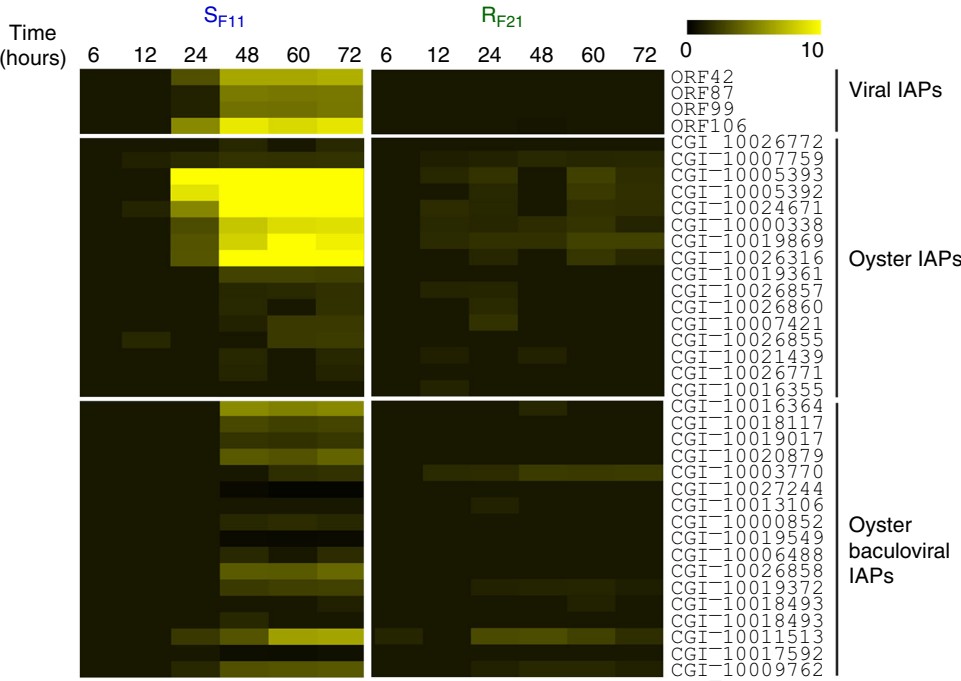

**Fig. 6** Viral and oyster endogenous inhibitors of apoptosis (IAPs) are strongly induced in the susceptible $S_{F11}$ oysters (but not in the resistant $R_{F21}$ oysters) during the 'natural' experimental infection. The fold changes in virus IAPs, oyster IAPs, and oyster and baculoviral IAPs were calculated between each time point of the kinetics and the T0. Analyses were conducted with the RNA-seq data through mapping against the *C. gigas* genome for oyster IAPs and against the OsHV-1 genome for viral IAPs[30]. The intensity of the colour indicates the magnitude of the differential expression (log2 fold change). The heatmap was constructed with Multiple Array Viewer software

Fig. 5c) were strongly enriched in $S_{F11}$ oysters. However, this intense antiviral response was inefficient, as the replication of the virus in $S_{F11}$ oysters was very high at these time points (Fig. 2a). Among the functional categories that were highly induced in $S_{F11}$ oysters only, we found the negative regulation of cell death category to be of particular interest (Supplementary Data 4). We performed a detailed analysis of the expression of the corresponding genes encoding endogenous inhibitors of apoptosis proteins (IAPs and Baculoviral IAPs, Fig. 6). These genes were found to be highly induced in $S_{F11}$ oysters, but only from 24 h to the end of the experimental infection. In addition, viral transcriptome analysis revealed an over-representation of viral transcripts encoding putative IAPs (ORFs 42, 87, 99 and 106) at the same time points in $S_{F11}$ oysters (Fig. 6). These data indicate that $S_{F11}$ oysters mount a delayed inefficient antiviral response and have an impacted apoptosis regulation during the late phase response.

**OsHV-1 infection alters the antibacterial defence of oysters**. Histological sections of $S_{F11}$ oysters showed accumulation of OsHV-1 in the haemocytes (Fig. 7), which are oyster immune cells that play a major role in controlling bacterial infections[33]. Interestingly, histological sections of $R_{F21}$ oysters were devoid of OsHV-1 positive cells (Supplementary Fig. 11). As a proxy to monitor the antibacterial defence of oysters, we analysed the expression of genes encoding antimicrobial peptides (AMPs) in both oyster families. AMPs showed highly contrasted expression patterns in $R_{F21}$ and $S_{F11}$ oysters during the experiment (Fig. 8). In resistant ($R_{F21}$) oysters, expression of *Cg*-BigDef1 and 2 was induced. Conversely, expression of *Cg*-BigDef1, *Cg*-BigDef3 and *Cg*-PRP transcripts decreased significantly and regularly over time in $S_{F11}$ oysters (Kruskal–Wallis, $p = 0.009$, 0.009, and 0.005, respectively). This overall repression of AMP expression, which

has not been previously reported, was particularly strong after 24 h, which corresponds to the period of microbiota destabilization preceding bacterial proliferation in $S_{F11}$ oysters (Figs. 2, 3 and 4). Interestingly, both *Cg*-BigDefs and *Cg*-PRPs are produced only by haemocytes[34]. We also showed that the transcript abundance of *Cg*-EcSOD, a specific marker of haemocytes[35], decreased over time in $S_{F11}$ oysters only (Fig. 8). Taken together, our data strongly suggest that by invading haemocytes of susceptible oysters, OsHV-1 infection alters haemocyte functions and thereby disrupts an important component of the oyster antibacterial shield.

**Similar events lead to similar phenotypes across families**. To determine whether the main molecular events observed in $S_{F11}$ and $R_{F21}$ oysters were conserved in other oyster families sharing the same phenotypes but with distinct genetic backgrounds, we monitored viral replication, bacterial load and gene expression in two additional susceptible families, $S_{F14}$ and $S_{F15}$, which died at more than 96% (Fig. 1), and in two other resistant families ($R_{F23}$ and $R_{F48}$), which died at less than 18% (Fig. 1). Consistent with our previous observations for $R_{F11}$ and $S_{F21}$, viral infection occurred early (12 h) after the beginning of the experimental infection in all oyster families, but OsHV-1 replicated intensely in susceptible families ($S_{F14}$ and $S_{F15}$) only (Fig. 9a). Subsequently, these two families were heavily colonized by bacteria (pairwise $t$-test at T72h; d.f. = 10; $p = 0.0027$), including vibrios and *V. crassostreae* (Fig. 9b-d). Like $S_{F11}$, the susceptible families $S_{F14}$ and $S_{F15}$ exhibited a high induction of both oyster and viral IAP (pairwise $t$-test at T72h; d.f. = 10; $p = 0.0025$ and $p < 0.0001$, respectively), together with lower expression or repression of haemocyte genes encoding AMPs and *Cg*-EcSOD (Fig. 9e, f). In contrast, oysters from resistant families $R_{F23}$ and $R_{F48}$ displayed neither such a massive induction of IAP nor a repression of

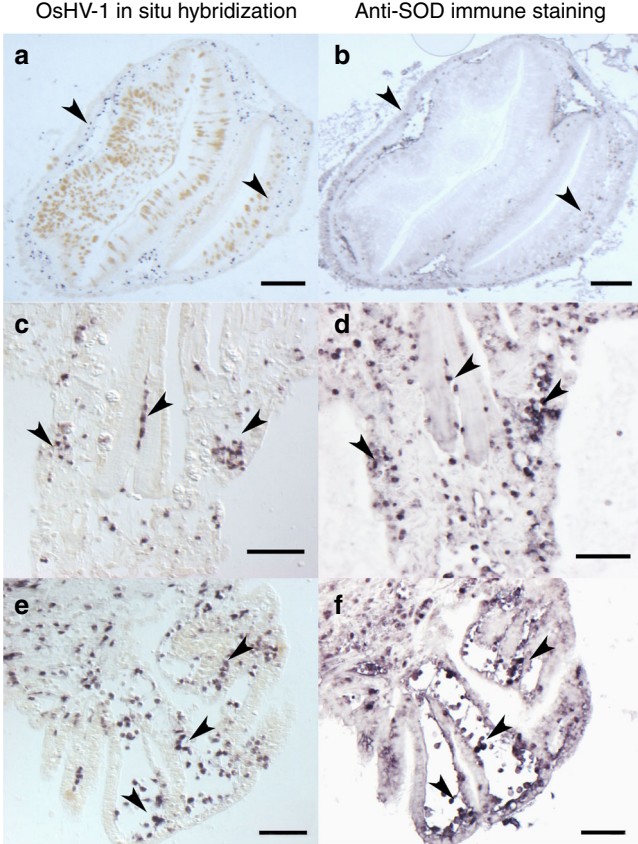

OsHV-1 in situ hybridization     Anti-SOD immune staining

**Fig. 7** OsHV-1 invades haemocytes of the susceptible S$_{F11}$ oysters. OsHV-1 was detected in S$_{F11}$ by in situ hybridization (**a**, **c**, **e**). Paraffin wax-embedded sections of oysters that were fixed 54 h after the beginning of the 'natural' experimental infection were hybridized with a specific probe labelled with digoxigenin and revealed using alkaline phosphatase-conjugated antibodies and NBT/BCIP (dark blue precipitate). Haemocytes were localized on consecutive tissue sections by immunostaining with an antibody specific to the SOD haemocytic protein (**b**, **d**, **f**). Immunostaining was revealed using alkaline phosphatase-conjugated secondary antibodies. Labelling of virus-infected cells was observed in areas enriched in haemocytes in different tissues (indicated by arrowheads), such as the connective tissue surrounding the digestive gland (**a**, **b**, scale bars 50 μM) or in the gills (**c**–**f**, scale bars 20 μM)

*Cg*-EcSOD or AMP expression, which were instead induced (Fig. 9e, f), confirming earlier results for the R$_{F21}$ family (Figs. 6 and 8). Finally, we showed that early induction of antiviral genes (Viperin, cGAS, IRF, TNF and SOCS2) at 6 h (pairwise *t*-test; d.f. = 10; $p = 0.1841$, $p = 0.0063$, $p = 0.0317$, $p = 0.0024$, $p = 0.0409$, respectively) and 12 h (pairwise *t*-test; d.f. = 10; $p = 0.0012$, $p = 0.0011$, $p = 0.0143$, $p = 0.0009$, $p < 0.0001$, respectively) was the hallmark of resistant families (Fig. 9g), as also evidenced by the RNA-seq data from R$_{F21}$ oysters (Supplementary Data 3).

**Viral infection and bacteraemia are needed to kill oysters.** To disentangle the respective roles of OsHV-1 and bacteria in pathogenesis, we designed a series of experimental infections using genetically diversified juveniles (Fig. 10). In a first set of experiments, we mimicked the simultaneous transmission of OsHV-1 and vibrios from infected to healthy oysters. Recipient (healthy) oysters were simultaneously exposed to two types of donors: the first ones were injected with *V. crassostreae* (pathogenic strain J2-9[21]), and the second ones were injected with OsHV-1 (Fig. 10a–j). The mortality of the donors is presented in Supplementary Fig. 12. Recipient oysters were either injected with poly(I:C), which restricts OsHV-1 replication[36], with sterile seawater (SW) as a control (Fig. 10a–e), or treated with chloramphenicol, a broad-spectrum bacteriostatic antibiotic that is used to limit oyster colonization by bacteria, particularly vibrios[12] (Fig. 10f–j). The mortality and pathogen load of recipient oysters were monitored throughout disease development. Injection of poly(I:C), as opposed to SW, was sufficient to completely block OsHV-1 replication (Fig. 10b), bacterial colonization (Fig. 10c, d and e) and the death of recipient oysters (Fig. 10a). Moreover, antibiotic treatment significantly reduced the load of vibrios (pairwise *t*-test at T72h; d.f. = 2; $p = 0.0009$, Fig. 10i) including *V. crassostreae* (pairwise *t*-test at T72h; d.f. = 2; $p = 0.028$, Fig. 10j) and oyster mortality (Mantel–Cox log-rank test, $p = 0.0075$, Fig. 10f) without affecting OsHV-1 replication (Fig. 10g) and the total bacterial load associated with oysters (Fig. 10h). Importantly, when we mimicked the transmission of only one pathogen (Fig. 10k–o), the mortality of recipient oysters was observed only when they were exposed to virus-injected donors (Fig. 10k). In this condition, viral replication (Fig. 10l) was accompanied by increases in the total bacterial (Fig. 10m) and total vibrio loads (Fig. 10n). Nevertheless, *V. crassostreae*, which was not included in this last experimental infection set-up, was not detected in oyster flesh (Fig. 10o). Two additional rationalized experimental infections were performed using the same design with susceptible juveniles from a biparental family (family H12[10]) (Supplementary Fig. 13). Both experiments confirmed the

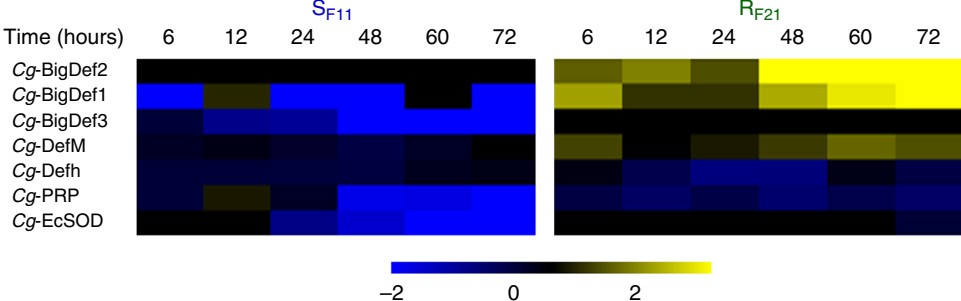

**Fig. 8** Antimicrobial peptides and *Cg*-EcSOD expression is repressed in the susceptible S$_{F11}$ oysters. Time-course of antimicrobial peptide (AMP) and *Cg*-EcSOD expression in S$_{F11}$ and R$_{F21}$ oysters during the 'natural' experimental infection. The relative expression of AMPs was analysed by comparing the number of reads (calculated by alignment using DIAMOND 0.7.9) between each time point and time zero. Analyses were performed using the RNA-seq data by mapping against the *C. gigas* genome for *Cg*-EcSOD. The intensity of the colour from blue to yellow indicates the magnitude of the differential expression (log2 fold change). The heatmap was constructed with Multiple Array Viewer software

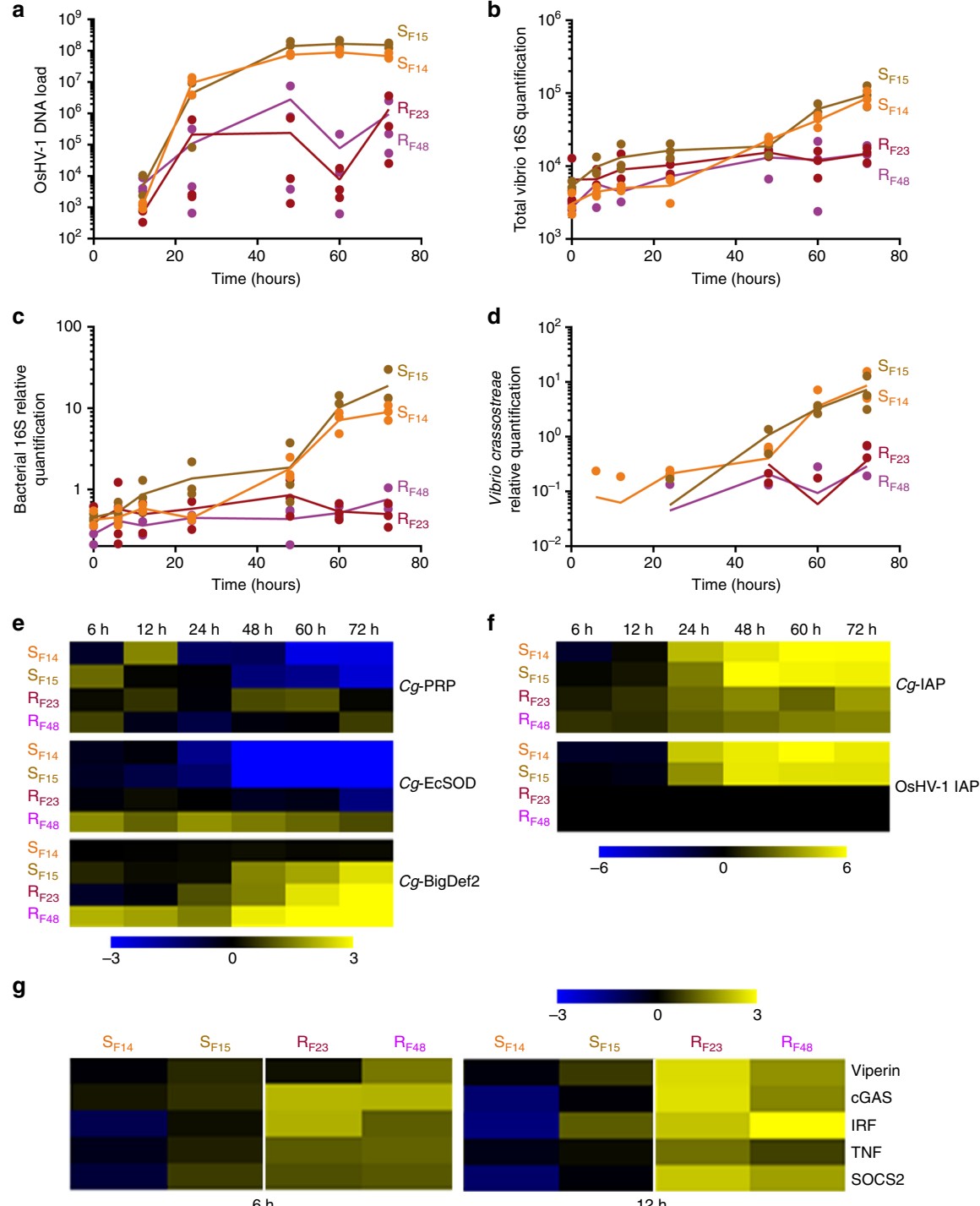

**Fig. 9** Conservation of molecular events during the 'natural' experimental infection in the susceptible $S_{F14}$ and $S_{F15}$, and in the resistant $R_{F23}$ and $R_{F48}$ oysters. **a** The OsHV-1 load was quantified by qPCR. Quantification of total *Vibrio* (**b**), total bacteria (**c**) and *Vibrio crassostreae* (**d**) were performed by qPCR. For **a** to **d**, dots represent distinct pools of 10 oysters. The mean values (plain lines) are displayed. Heatmap of antimicrobial peptide (*Cg*-Bigdef2, *Cg*-PRP), *Cg*-EcSOD (**e**) and IAPs (**f**) expression measured by RT-qPCR in each oyster family. **g** Heatmap of antiviral gene (Viperin, cGAS, IRF, TNF and SOCS2) expression measured by RT-qPCR in each oyster family at early time points (6 and 12 h) of the experimental infection. The intensity of the colour from blue to yellow indicates the magnitude of the differential expression (log2 fold change). The heatmap was constructed with Multiple Array Viewer software

essential role of OsHV-1 replication in the bacterial colonization and death of recipient oysters (mortality data for donors are provided in Supplementary Fig. 14).

To identify the bacterial species involved in the secondary bacterial infection in our rationalized experiments, we studied

microbiota changes in recipient oysters using 16S rDNA metabarcoding (Supplementary Data 5). All bacterial genera that changed significantly during these experimental infections are reported in Supplementary Data 6. Among them, the most abundant in oysters (more than 4% of the total bacterial

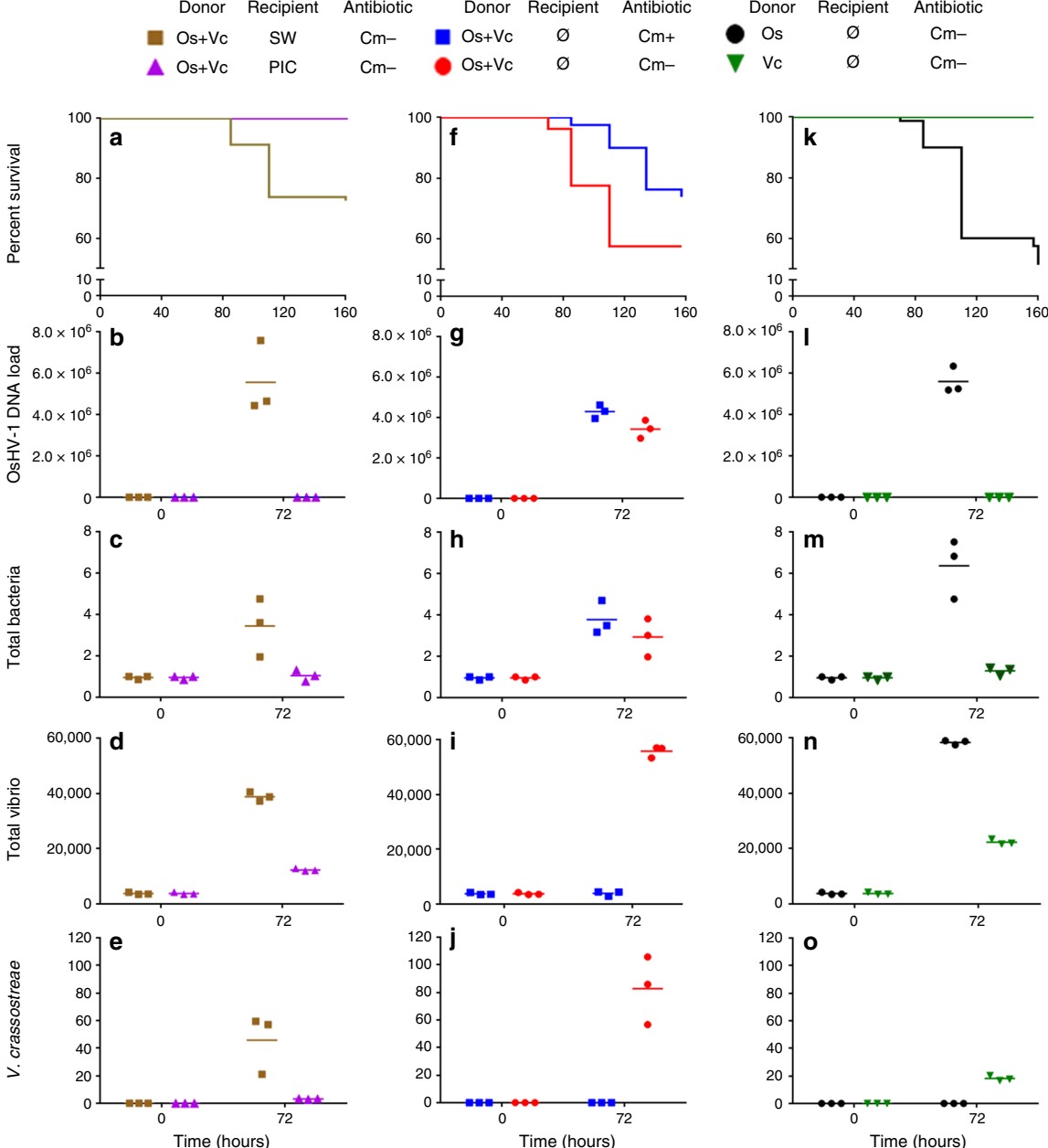

**Fig. 10** Rationalized experimental infections demonstrate that OsHV-1 replication is required for bacterial colonization and oyster death. Experimental infection by OsHV-1 and/or *V. crassostreae* was performed as follows: oyster donors were injected with either $3.88 \times 10^8$ genomic units of OsHV-1 (Os) or $5 \times 10^7$ cfu *V. crassostreae* (Vc). **a–e** Recipient oysters were injected with poly(I:C) (PIC) or sterile seawater (SW) before exposition to both Vc and Os donors. **f–j** Recipients were exposed to both Os and Vc donors in the presence (Cm + ) or absence (Cm−) of chloramphenicol in the tanks. **k–o** Recipients were exposed to Os or Vc donors. The OsHV-1 DNA load (viral genomic units per ng of total oyster DNA), relative quantification of total bacteria, and total Vibrio and *Vibrio crassostreae* abundance were measured by qPCR. No mortality was observed, and no OsHV-1 DNA was detected in recipient oysters when untreated donors were used as a control. Each dot represents a measure done on a distinct pool of 10 oysters

communities in at least one sample) (Supplementary Fig. 15) included OTUs from different genera, which increased significantly in the four treatments in which mortalities occurred (Supplementary Fig. 15b, c, d and e). Two of them (*Vibrio* and *Arcobacter*) were common to three out of four treatments (Supplementary Fig. 15b, c and e). These two genera were also found to colonize the recipient oysters in our initial 'natural' experimental infection (Fig. 3b). Finally, although chloramphenicol treatment reduced and delayed mortality (Fig. 10b), secondary bacterial infection could still occur involving bacteria from other genera (Supplementary Fig. 15d). Taken together,

these results indicate that viral infection is necessary to initiate the infectious process and that opportunistic bacteria are responsible for a secondary infection necessary to complete disease development.

## Discussion

In this study, we deciphered the complex intra-host interactions underlying the mortality syndrome affecting juveniles of *Crassostrea gigas*, providing a comprehensive view of the pathogenic processes underpinning a disease that has remained

incompletely understood until now. The entire sequence of events leading to oyster death was traced and identified.

We showed that infection by the OsHV-1 µVar is the first event that occurs during the infectious process and that the intense replication of this virus is a prerequisite for development of the disease. We showed that the immune cells of oysters, the haemocytes, are one of the cell types targeted by the virus. This localization of OsHV-1 in haemocytes during the infectious process has also been recently reported[37]. Notably, infection of haemocytes by OsHV-1 impacted haemocyte physiology and particularly impaired the expression of AMPs either through transcriptional regulation or indirectly through the induction of cell death or lysis processes, as previously reported[37]. Following the repression of antibacterial defences, profound changes in oyster-associated microbiota were observed, followed by bacteraemia and mortalities. These results clearly identify OsHV-1 as a necessary actor triggering the disease as a whole.

Bacterial colonization was shown to be the second event necessary to complete the infectious process leading to death in oysters. This finding supports previous studies identifying bacteria as important aetiological agents of the disease[12]. From our histological analysis, gills could be the initial route of bacterial infection in OsHV-1 immune-compromised oysters before bacterial dissemination to the rest of the tissues. Such a scenario is reminiscent of immune suppression by viruses such as HIV and secondary colonization by opportunists[38]. Bacteria of the *Vibrio* genus were associated with the disease, supporting previous studies that identified *V. crassostreae* as an important pathogenic population in the field[21,22]. However, (i) *V. crassostreae* was not required to complete the infectious process leading to oyster death, and (ii) vibrios were not the only bacteria systematically associated with dying oysters. Finally, although chloramphenicol treatments controlling vibrios reduced and delayed mortality, secondary bacterial infection could still occur involving bacteria from other genera. These results clearly showed that the secondary bacterial infection of oysters already immune-compromised by viral infection could be engaged by a series of opportunistic bacteria present in the environment.

The inability of susceptible oysters to control viral replication and further bacterial colonization was associated with a strong, but late, antiviral response. Importantly, the molecular function negative regulation of cell death was highly mobilized concomitantly with the intense replication of the virus in susceptible oysters. The majority of the genes belonging to this function encode endogenous IAPs that were strongly induced in susceptible oysters. Remarkably, intense OsHV-1 replication was also associated with high expression of IAPs of viral origin. Such viral proteins of the BIR family are known to have anti-apoptotic activities favouring viral replication[39]. These results indicate that both endogenous and exogenous anti-apoptotic processes, which are strongly activated in susceptible oysters, may play a key role in the success of OsHV-1 infection[26,40].

Oyster resistance was associated with an early limitation of viral replication during pathogenesis. Although the genomic determinants of the antiviral response of resistant oysters remain to be identified and validated functionally, antiviral pathways that are known to be necessary for resistance to herpesviruses in vertebrates[41] were shown to be highly induced at early times in resistant oysters only. Genes involved in these pathways could be valuable candidates for future selective breeding. Overall, our data indicate that the time required for an oyster to establish effective immunological control is a key indicator of disease outcome. This finding is in agreement with theoretical predictions indicating that variations in parameters such as pathogen expansion or host response dynamics can affect the outcome of the infection[42]. Recent studies in an insect model of bacterial infection also support this theoretical prediction and identified the early induction of AMP expression as a key determinant of an efficient response against infection[43].

In conclusion, the present work enabled us to decipher the mechanisms underlying a complex pathosystem affecting juvenile oysters. We found that pathogenesis was caused by multiple infections involving a virus and opportunistic bacteria. Indirect intra-host interactions were shown to occur during pathogenesis, enabling bacterial colonization of oysters that were already immune-compromised by the virus. As in a suprainfection pattern[44], we found that the bacteria could not infect oysters in the absence of the virus, and we never observed oysters infected by only one of the pathogens. Future studies are needed to validate this suprainfection model and explore the genetic and physiological attributes underlying initial and subsequent colonization waves in oysters. Characterization of this multifactorial disease represents a breakthrough that was made possible by holistic approaches developed by combining 'natural' experimental procedures, using oyster biparental families with contrasted susceptibilities to the disease and developing thorough molecular analyses of host responses, the microbiota structure and pathogen monitoring. This holistic view of diseases as a system provides an exceptionally well-adapted framework for studying the factors governing the progression of infections. We believe that such an integrative and holistic approach could now be applied to a series of multifactorial diseases that affect non-model invertebrate species worldwide.

## Methods

**Production of biparental oyster families.** In 2015, fifty different biparental *Crassostrea gigas* oyster families were produced from wild seed broodstocks sampled in farming and non-farming areas in two geographic regions (French Mediterranean and Atlantic coasts, Supplementary Table 1). In the Atlantic area, 73 oysters were collected at Logonna Daoulas (farming area) and 70 oysters at Dellec (non-farming area). In the Mediterranean area, 125 oysters were collected in the Thau lagoon (farming area) and 65 at the Vidourle river mouth (non-farming area). In addition, 84 oysters issued from a mass selection programme to enhance their resistance to mortality syndrome were used[28]. All the collected oysters were transferred to the Ifremer facility at Argenton (Brittany, France) between 6 and 8 January 2015 and treated for 6 days with chloramphenicol (8 mg/l).

For gametogenesis induction, animals were held for 8 weeks in 500 l flow-through tanks with seawater enriched with a phytoplankton mixture at a constant temperature of 17 °C[13,22]. Seawater was UV-treated and filtered through 10-µm mesh. The daily mixed diet consisted of *Tisochrysis lutea* (CCAP 927/14; 40 µm³, 12 pg cell⁻¹) and *Chaetoceros muelleri* (CCAP 1010/3; 80 µm³, 25 pg cell⁻¹). Once the oysters were reproductively mature, gametes from 91 individuals (46 males, 45 females) were obtained by stripping. Gametes from one male and one female from the same origin were mixed in a 5-l cylinder at a ratio of 50 spermatozoids per oocyte (day 0). The fertilized oocytes completed their embryonic development in 5-l tubes filled with 1-µm-filtered, UV-treated seawater at 21 °C for 48 h. The D-larvae (day 2) were then collected and reared in flow-through rearing systems at 25 °C[45]. At the end of the pelagic phase (day 15), all the larvae were collected on a 100-µm sieve and allowed to settle on culch. Post-larvae were maintained in downwelling systems, where they were continuously supplied with enriched seawater until the experiments began. In the larval and post-larval stages, the oysters were fed the same diet as the broodstock at a concentration between 1500–2000 µm³ µl⁻¹[45].

Of the 50 families of oyster seed produced, 3 families from each location were kept, along with 3 other families from the mass selected broodstock, for 'natural' experimental infections. These 15 oyster families were maintained under highly controlled biosecured conditions to be sure that no oyster pathogens would interfere with further experiments. The 'pathogen-free' status of the animals was confirmed by (i) the absence of OsHV-1 DNA detection by qPCR and (ii) a low *Vibrio* presence (~10 cfu⁻¹ tissue) determined by isolation on selective culture medium (thiosulfate-citrate-bile salts-sucrose agar, TCBS)[12]. Oysters were observed to remain free of any abnormal mortality throughout the larvae until the beginning of the 'natural' experimental infections.

**'Natural' experimental infections.** Our experimental infection protocol consists of a cohabitation between *C. gigas* oysters ('donors') carrying the disease and 'pathogen-free' *C. gigas* oysters ('recipients')[13,22]. 'Pathogen-free' oysters used as donors (a mixture of 116-day-old oysters from 15 families, 17,700 g with a mean individual weight of 1.1 g corresponding to a weight of flesh without shell of ~0.2 g) were first deployed in a farming area (Logonna Daoulas, (lat 48.335263—long

−4.317922) during the infectious period until the first mortalities occurred (0.01% on 17 July 2015). Then, donor oysters were transferred back to the laboratory and placed in contact with 'pathogen-free' recipient oysters in a controlled environment[11–13,22] (Supplementary Fig. 1). The experiment was conducted by placing the same biomass (1120 g) of donors in cohabitation in 15 independent tanks (500 l), with each containing one of the 15 families (recipient oysters with a mean individual weight of 1.1 g corresponding to a weight of flesh without shell of ~0.2 g) acclimatized in these structures for 2 weeks. In parallel, a control cohabitation experiment was performed under identical conditions but using donors that had not spent time in the farming areas. The 'natural' experimental infection began on 17 July 2015 and ended on 31 July 2015. Mortality was monitored in laboratory tanks. When recipients were exposed to the donors (17 July 2015), 2 replicates of 100 'pathogen-free' oysters of each family were placed in the farming area, and mortality was monitored daily.

During the experimental infection, 10 oysters in triplicate were randomly sampled without blinding protocols from each tank and at each time (0, 6 h, 12 h, 24 h, 48 h, 60 h and 72 h) of the kinetics. The shell was removed, and pools of 10 oysters were flash frozen in liquid nitrogen. Oyster pools (10 individuals per pool) were ground in liquid nitrogen in 50-ml stainless steel bowls with 20-mm-diameter grinding balls (Retsch MM400 mill). The obtained powders (stored at −80 °C) were then used for extraction of RNA and DNA. In addition, 5 oysters were sampled at 54 h and 78 h and fixed in Davidson fixative[46] for histological analyses.

**Rationalized experimental infections.** The experiments were performed using genetically diversified *C. gigas* oysters (4 months old with a mean individual whole weight of 1 g corresponding to a weight of flesh without shell of ~0.2 g) or susceptible biparental family H12[10] *C. gigas* oysters (4 months old with a mean individual whole weight of 1.5 g corresponding to a weight of flesh without shell of ~0.3 g). Oysters from these genetic backgrounds were 'pathogen-free' and were produced as previously described[13]. For the experiments, the oysters were anesthetized in hexahydrate MgCl₂ (ACROS, 50 g l⁻¹, 100 oysters l⁻¹) for 2 h[47]. Then, they were injected using a 26-gauge needle attached to a multi-dispensing hand pipette in the adductor muscle to allow spreading into the circulatory system with either 20 µl viral or 40 µl bacterial inoculum. The OsHV-1 inoculum (3.88 × 10⁸ OsHV-1 genomic units µl⁻¹) was prepared according to previously described protocols[48], and injections were performed 24 h before the start of the experiment. *Vibrio crassostreae* J2-9 was grown under agitation at 20 °C in Luria-Bertani (LB) + NaCl 0.5 M for 18 h[21]. The culture was centrifuged (1000× g, 10 min, 20 °C), suspended in culture medium to an optical density (OD₆₀₀) of 1 and injected (5 × 10⁷ CFU) immediately before the start of the experiment. The rationalized experimental infections were performed by exposing the recipients (n = 100) to injected donors (n = 100) in 50-l tanks at 21 °C. A control experiment was carried out by placing the non-injected donors in contact with recipient oysters. During each experimental treatment, oysters were sampled at 0 and 72 h, and cumulative mortality was monitored up to 72 h and 157 h for both donors and recipients, respectively. Recipient oyster pools were ground in liquid nitrogen in 50-ml stainless steel bowls with 20-mm-diameter grinding balls (Retsch MM400 mill). The powders obtained (stored at −80 °C) were then used for DNA extraction. In one of these rationalized experimental infections, chloramphenicol (8 mg l⁻¹) was added to the tank every 2 days to restrict bacterial proliferation. In another experiment, the recipients were treated with poly(I:C), which has been shown to block OsHV-1 replication[36]. In this case, 20 µl poly(I:C) (HMW, InvivoGen, cat. code: tlrl-pic - 1 mg ml⁻¹) was injected into the adductor muscle of recipient oysters 24 h before the start of the cohabitation experiment. The experiment with genetically diversified oysters was performed one time. The experiments using oysters from the H12 family were replicated two times. Three pools of 10 individuals for each experiment were randomly sampled without blinding protocols, and submitted for further molecular analysis.

**Oyster transcriptome analyses.** RNA was extracted from the powdered oysters using the Direct-Zol RNA Miniprep kit (Proteigene) according to the manufacturer's protocol. RNA concentration and purity were checked using a Nanodrop ND-1000 spectrometer (Thermo Scientific), and their integrity was analysed by capillary electrophoresis on a BioAnalyzer 2100 (Agilent). RNA-seq library construction and sequencing were performed by the Fasteris Company (Switzerland). Directional cDNA libraries were constructed using a TruSeq mRNA Stranded kit (Illumina) and sequenced on a Hiseq in paired-end reads of 2 × 75 bp. All data treatments were carried out under a local galaxy instance[49]. Phred scores were checked using the Fastq-X toolkit[50] and were higher than 26 over 90% of the read length for all the sequences. All the reads were thus kept for subsequent analyses. Mapping to the *C. gigas* reference genome (assembly version V9[51]) was performed using RNAstar (Galaxy Version 2.4.0d-2[52]). The HTSeq-count was used to count the number of reads overlapping annotated genes (mode Union) (Galaxy Version v0.6.1)[53]. Finally, the differential gene expression levels were analysed with the DESeq2 R package[54]. Fold changes between each time point of the kinetics and the T0 control condition were considered significant when the adjusted p-value (Padj) for multiple testing with the Benjamini–Hochberg procedure, which controls the false discovery rate (FDR), was < 0.05.

Because not all known *C. gigas* AMPs were present in the *C. gigas* reference genome (assembly version V9[51]), read counts for all of them were specifically

obtained by alignment against a protein database using DIAMOND 0.7.9[55]. The AMP database was prepared by retrieving *C. gigas* sequences from GenBank that were manually inspected to discard irrelevant or incomplete sequences. The reads for each sample/replicate were compared with the database using DIAMOND blastx. Alignments were filtered for the best hit and e-value < 1e-6. Read counts for each AMP were normalized against both the transcript size and the total sequence number for each sample/replicate and used for the differential gene expression analysis with DESeq2.

**Gene ontology annotation and enrichment analysis.** To work with current functional annotations of the *C. gigas* gene set, we performed a de novo functional annotation. Blastx comparison against the NR database was performed for the 28,027 genes annotated in the genome, with a maximum number of target hits of 20 and a minimum e-value of 0.001. XML blast result files were loaded onto Blast2GO[56] for GO mapping and annotation with the b2g_sep13 version of the B2G database. These results were used as inputs for GO enrichment analysis, which was performed using adaptive clustering and a rank-based statistical test (Mann–Whitney U-test combined with adaptive clustering). The R and Perl scripts used[57] can be downloaded [https://github.com/z0on/GO_MWU]. The following parameters were used for adaptive clustering: largest = 0.5; smallest = 10; clusterCutHeight = 0.25. For the continuous value characterization of each gene in the data set, we used a strategy aiming to take into account both the level of expression and the significance of the differential expression. To combine these two factors, the log2 fold change was attributed to genes that were significantly differentially expressed (adjusted p < 0.05), while a zero was attributed to the others. A category was considered enriched with a FDR < 1%. To represent the results synthetically, the intensity of the enrichment was calculated with the following ratios: i) for the upregulated enriched categories, 'number of genes significantly upregulated in the category/total number of genes in the category'; (ii) for the downregulated enriched categories, '−1 × (number of genes significantly downregulated in the category /total number of genes in the category)'. These ratios were then displayed on a heatmap using the Multiple Experiment Viewer and clustered according to the Pearson correlation[58].

**Virus transcriptome analyses.** RNA-seq reads that did not align with the *C. gigas* genome were collected using SAMtools[59]. These reads were aligned with the viral genome sequence of OsHV-1 (Refseq NC_005881.2[30]) using bowtie2 with single-end global alignment and default parameters[60]. The abundance of viral transcripts was calculated using HTSeq-count[53] and the viral genome GFF3 file. The counts for each ORF were normalized by dividing by the size of the ORF and multiplying by a library normalization factor (calculated as the average library size for all times and controls divided by the library size for the specific time point). $C = \left(\frac{c_{orf}}{s_{orf}}\right) \times \left(\frac{\bar{L}}{L_i}\right)$, where $c_{orf}$ represents the raw counts for the ORF, $s_{orf}$ represents the size of the ORF, $\bar{L}$ represents the average reads for all the time points and $L_i$ corresponds to the number of total reads for the specific time point.

**Quantification of bacteria and viruses.** Quantification of OsHV-1, total 16S rDNA Vibrio and *Vibrio crassostreae* was performed using quantitative PCR (qPCR). All amplification reactions were analysed using a Roche LightCycler 480 Real-Time thermocycler (qPHD-Montpellier GenomiX platform, Montpellier University, France). The total qPCR reaction volume was 1.5 µl and consisted of 0.5 µl DNA (40 ng µl⁻¹) and 1 µl LightCycler 480 SYBR Green I Master mix (Roche) containing 0.5 µM PCR primer (Eurogenetec SA). Virus-specific primer pairs targeted a region of the OsHV-1 genome predicted to encode a DNA polymerase catalytic subunit (ORF100, AY509253): Fw-ATTGATGATGTGGATAAT CTGTG and Rev-GGTAAATACCATTGGTCTTGTTCC[30]. Total bacteria specific primer pairs were the 341F-CCTACGGGNGGCWGCAG and 805R-GACTACHV GGGTATCTAATCC primers targeting the variable V3V4 loops for bacterial communities[61]. Total Vibrio specific primer pairs were Fw-GGCGTAAAGCGCA TGCAGGT and Rev-GAAATTCTACCCCCCTCTACAG[62], and *Vibrio crassostreae*-specific primer pairs were Fw-ATGACCATCCAACAACCCG and Rev-AGC CGTAATTGATACGCACG. A Labcyte Acoustic Automated Liquid Handling Platform (ECHO) was used for pipetting into the 384-well plate (Roche). A LightCycler® 480 Instrument (Roche) was used for qPCR with the following program: enzyme activation at 95 °C for 10 min, followed by 40 cycles of denaturation (95 °C, 10 s), hybridization (60 °C, 20 s) and elongation (72 °C, 25 s). A subsequent melting temperature curve of the amplicon was performed to verify the specificity of the amplification. Absolute quantification of viral and bacterial DNA copies were estimated by comparing the observed Cq values to a standard curve of the DP amplification product cloned into the pCR4-TOPO vector for OsHV-1 and from total DNA extraction of *V. crassostreae* J2-9 for total vibrio 16S rDNA. For total bacteria and *V. crassostreae* 16S rDNA, we used the relative quantification calculated by the 2^(−ΔΔCq) method[63] with the mean of the measured threshold cycle values of two reference genes (Cg-BPI, GenBank: AY165040 and Cg-actin, GenBank: AF026063).

**Giemsa staining and in situ hybridization.** *Crassostrea gigas* tissues were fixed for 24 h in Davidson fixative[46]. Tissues were embedded in paraffin wax, serially sectioned to a thickness of 5 µm and collected on polylysine-coated slides (performed

by Histalim Company, France). To visualize bacteria that had infiltrated the tissues, tissue sections were stained using Giemsa, which coloured the different tissues and cells in shades of pink to purple and most bacteria in deep blue (performed by Histalim Company, France). The presence of OsHV-1 in tissue sections was detected by in situ hybridization following a previously published protocol[64]. The slides were hybridized with 5 ng μl$^{-1}$ of OsHV-1-specific digoxigenin-labelled (DIG) antisense probes designed based on the C2-C6 fragment of the ORF4 of the OsHV-1 reference genome (GenBank: NC_005881.2), which is also present in the OsHV-1 μVar sequence. As a negative control, for the specificity assessment of the OsHV-1 probe, the samples were also hybridized with a GFP probe with no homology to the OsHV-1 or C. gigas genomes. The primers used for the probe synthesis were as follows: OsHV-1 probe, C2-CTCTTTACCATGAAGATACCCACC and C6-GTGCACGGCTTACCATTTTT; GFP probe, GFP-Fw-ACGTAAACGGCCACAAGTTC and GFP-Rev-AAGTCGT GCTGCTTCATGTG. After hybridization, the tissues were counter-stained with a solution of Bismark Brown yellow. Haemocyte localization in tissue sections was performed by immunohistology. After dewaxing the tissue sections in xylene followed by rehydration in an ethanol series and distilled water, a heat-induced antigen retrieval procedure was conducted by incubating the sections for 12 min in sodium citrate solution (10 mM, pH 6) in a microwave (800 W). The sections were then incubated at room temperature for 1 h in a 5% non-fat dry milk (NFDM) and 0.1% Triton X100 solution as a blocking agent and permeabilization solution. Immunodetection of the haemocyte-specific protein Cg-EcSOD was performed using a primary antibody produced in-house from mouse ascites[35]. Sections were incubated overnight in a humidified chamber at 4 °C in a 1/500 dilution (in PBS-5% NFDM) of the primary antibody and rinsed 3 × 5 min in TBS (pH 7.4). A 1/5000 dilution of the secondary antibody (goat anti-mouse polyvalent immunoglobulin alkaline phosphatase-conjugated, SIGMA A0162) was then applied for 2 h at room temperature in a humidified chamber, and the sections were rinsed 2×5 min in TBS. Alkaline phosphatase enzymatic activity was revealed by incubating slides in NBT/BCIP solution for 20 min in the dark, and reactions were stopped by thoroughly rinsing the slides in distilled water. As a negative control for specificity assessment of the anti-Cg-EcSOD signal, the sections were incubated with the secondary antibody only. The slides were finally mounted in Dako mounting medium (DAKO S3023). Images were acquired using a Zeiss microscope equipped with a Zeiss colour camera and managed with ZEN software (Montpellier RIO imaging platform). For in situ hybridization, labelling was observed only with the virus-specific probe, and no labelling was detected in the sections treated with the GFP probe (Supplementary Fig. 16). For Cg-EcSOD immunostaining, labelling was observed in sections treated with anti-Cg-EcSOD antibody, and no labelling was observed for slides treated with the secondary antibody only (Supplementary Fig. 17).

**RNA extraction and RT-qPCR analysis**. RNA extraction was performed using Direct-zol$^{\text{tm}}$ RNA MiniPrep according to manufacturer's instructions (Zymo Research). Frozen oyster powder (20 mg) was homogenized in 1 ml TRIzol by vortexing 1 h at 4 °C. Prior to extraction, insoluble materials were removed by centrifugation at 12,000× g for 10 min at 4 °C. The quantification and integrity of the total RNA were checked using a NanoDrop spectrophotometer (Thermo Scientific) and 1.5% agarose gel electrophoresis, respectively. Total RNA (300 ng) was reverse-transcribed in 20 μl using Moloney Murine Leukaemia Virus Reverse Transcriptase (MMLV-RT) according to the manufacturer's instructions (Invitrogen). Amplification and pipetting were performed with a Roche LightCycler 480 and a Labcyte Acoustic Automated Liquid Handling Platform (ECHO), as previously described (see above, Quantification of bacteria and viruses). The total RT-qPCR reaction volume was 1.5 μl and consisted of 0.5 μl of cDNA (dilution 1/6) and 1 μl of LightCycler 480 SYBR Green I Master mix (Roche) containing 0.5 μM of PCR primer (Eurogenetec SA). The amplification efficiency of each primer pair (Supplementary Tables 2 and 3) was validated by serial dilution of a pool of all cDNAs. Related expression was calculated as the threshold cycle (Cq) values of selected genes minus the mean of the measured threshold cycle (Cq) values of three constitutively expressed genes (Cg-EF1, Cg-RPL40 and Cg-RPS6).

**Bacterial microbiota analysis**. DNA from powdered oyster tissues was extracted with DNA from the tissue Macherey-Nagel kit (reference 740952.250) according to the manufacturer's protocol. Prior to 90 min of enzymatic lysis in the presence of proteinase K, an additional 12-min mechanical lysis (Retsch MM400 mill) was performed with zirconia/silica beads (BioSpec). DNA concentration and purity were checked with a Nanodrop ND-1000 spectrometer (Thermo Scientific). For each sample, 16S rDNA amplicon libraries were generated using the 341F-CCTACGGGNGGCWGCAG and 805R-GACTACHVGGGTATCTAATCC primers targeting the variable V3V4 loops for bacterial communities[61]. Paired-end sequencing with a 250-bp read length was performed at McGill University (Génome Québec Innovation Centre, Montréal, Canada) on a MiSeq system (Illumina) using v2 chemistry according to the manufacturer's protocol. The FROGS pipeline (Find Rapidly OTU with Galaxy Solution) implemented on a galaxy instance [http://sigenae-workbench.toulouse.inra.fr/galaxy/] was used for data processing[65]. In brief, paired reads were merged using FLASH[66]. After denoising and primer/adapter removal with cutadapt[67], clustering was performed using SWARM, which uses a novel clustering algorithm with a threshold (distance = 3) corresponding to

the maximum number of differences between two OTUs[68]. Chimeras were removed using VSEARCH[69]. We filtered out the data set for singletons and performed an affiliation using Blast + against the Silva 16S rDNA database (release 128, Sept 2016) to produce an OTU and affiliation table in the standard BIOM format. Rarefaction curves of species richness were produced using the R package and the rarefy_even_depth and ggrare functions[70]. We used phyloseq to obtain relative abundances at different taxonomic ranks (from genus to phylum) (tax_glom function). In our analyses, we only kept taxa that had a true annotation for each corresponding taxonomic rank (from genus to phylum).

**Statistical analyses**. Statistical analyses were performed using R v3.3.1 (R: a language and environment for statistical computing, 2008; R Development Core Team, R Foundation for Statistical Computing, Vienna, Austria [http://www.R-project.org]). Survival curves were used to analyse mortality kinetics, and the non-parametric Kaplan–Meier test was used to estimate log-rank values for comparing conditions. Principal coordinate analyses (PCoA, {phyloseq}) were computed to represent dissimilarities between samples using the Bray–Curtis distance matrix (ordinate, {phyloseq}). Multivariate homogeneity of group dispersions was tested between bacterial assemblages of $S_{F11}$ and $R_{F21}$ using 999 permutations (permutest, betadisper, {vegan}). To identify candidate taxa with changes in abundances between the initial and the final time points of the experiment, we used DESeq2 (DESeq[54]) from the OTUs to the higher taxonomic ranks. Heatmaps of significant genera were computed using relative abundances and the heatmap.2 function ({gplots}). We performed one-way ANOVA or non-parametric Kruskal–Wallis tests when the normality of residuals was rejected (Shapiro test) to compare alpha diversity metrics for bacterial microbiota, along with OsHV-1 (RNA and DNA level) and bacterial absolute abundances over the experiment after logarithmic transformations. When the ANOVA or Kruskal–Wallis tests were significant, we then computed pairwise comparisons between group levels (post-hoc analyses) with Bonferroni corrections for multiple testing using the pairwise t-test and the Dunn test, respectively. For all analyses, the threshold significance level was set at 0.05.

## Data availability

RNA-seq data and amplicon sequences for microbiota analysis have been made available through the SRA database (BioProject accession number PRJNA423079 with SRA accession SRP130264). RNA-seq data are available under SRA accessions SRR6679052-SRR6679093. Amplicon sequences for microbiota analysis during 'natural' experimental infections are available under SRA accessions SRR7786101–SRR7786142. Amplicon sequences for microbiota analysis during rationalized experimental infections are available under SRA accessions SRR6675783–SRR6675803. Other data analyzed during this study are included in this published article and its supplementary information files. Complementary information is available from the corresponding authors on reasonable request.

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

## Acknowledgements

We warmly thank the staff of the Ifremer stations of Argenton (LPI, PFOM) and Sète (LER), and the Comité Régional de Conchyliculture de Méditerranée (CRCM) for technical support in the collection of the oyster genitors and reproduction. We also thank Fabrice Pernet, Marie-Agnès Travers, David Mouillot, Marion Richard and Franck Lagarde for fruitful discussions. The authors are grateful to Philippe Clair from the qPHD platform/Montpellier genomix for useful advice and Yannick Labreuche for the *Vibrio crassostreae*-specific primers. This work, through the use of the GENSEQ platform [http://www.labex-cemeb.org/fr/genomique-envir-onnementale-2] from the labEx CeMEB, benefited from the support of the National Research Agency under the 'Investissements d'avenir' program (reference ANR-10-LABX-04-01).The authors also thank the Montpellier RIO imaging platform (https://www.mri.cnrs.fr). The present study was supported by the ANR project DECIPHER (ANR-14-CE19-0023), by the EU funded project VIVALDI (H2020 program, n°678589) and by Ifremer, CNRS, Université de Montpellier and Université de Perpignan via Domitia.

## Author contributions

J.D.L, A.L., B.P., C.M., J.-M.E, P.H., L.D., M.L., A.V., N.F., T.R., M.A.L., A.P., D.R., B.M., M.A.B., Y.G. and G.M. performed oyster experiments. J.D.L., A.L., E.T., C.C.L., M.C., Y.G. and G.M. performed microbiota analyses. J.D.L., A.L., E.T., C.M., J.V.D., C.C. H., R.G., J-.M.E., Y.G. and G.M. performed RNA-seq analyses. J.D.L., A.L., A.D., A.V. and C.M. performed qPCR analyses. C.M., G.M.C. and A.V. performed histology analyses. J.D.L., B.P., P.B., D.D-.G. and G.M. designed experiments. J.D.L., A.L., B.P., F.L.R., D.D-.G., Y.G. and G.M. interpreted results. J.D.L., A.L., F.L.R., D.D-.G., Y.G. and G.M. wrote the paper. All the authors have revised and approved the manuscript submission.

## Additional information

**Competing interests:** The authors declare no competing interests.

