## [Peer Review File · Nature Communications]

Reviewers' comments:

Reviewer #1 (Remarks to the Author):

Pacific oyster mortality syndrome is a disease of major significance to Pacific oyster culture globally. The authors use a multi-pronged approach to study the disease in resistant and susceptible seed Pacific oyster families to study the course of POMS. The major claim of the MS is that OsHV-1 is the primary causative agent but secondary bacterial infections are an important contributing factor in mortalities. One of the biggest strengths is the combination of experimental biology with both classic (histology) and modern tools (i.e. next generation sequencing, 16S tagged sequencing, ISH) focused on both the host and pathogens. I believe this MS will influence other researchers in the field, but has major limitations, i.e. the lack of replication in experimental approaches and pooling of animals for molecular analyses. I appreciate the use of both 'natural' and 'rationalized' experimental approaches, though both approaches lack replication, which concerns me about repeatability. I am also concerned with pooling animals for molecular analyses. If the oysters are large enough to inject, achieving high RNA quality from individuals should not be an issue; in fact pooling may include individuals that are not of high enough quality. The statistical analyses appear complete and appropriate for the data, except further data might be interesting to show within the supplementary files, in particular the fold change of genes of interest for the oyster—I'm curious to the underlying fold change differences (beyond the enrichment analysis) and also a heatmap of the viral gene expression over time. The authors use two experiment types; it would be helpful to the reader to define the experiment types as separate headings in the results and/or on the figures.

Specific line by line comments:

Line 71-74: The authors need to take care in how they describe OsHV-1 uvar. We don't currently have a good understanding of the number of microvariant strains globally; additionally the OIE only defines a specific type strain as OsHV-1 uvar.

Line 93-97: I suggest defining POMS much earlier in the introduction and provide a citation for the use of this acronym.

Line 133-135: How did you define this as OsHV-1 uvar?? Please include the specific modifications of the genome that defined this variant as OsHV-1 uvar. A table in the supplement describing the modifications would be helpful.

Line 135-137: I would suggest the following modification "Taken together, these results indicate a successful OsHV-1 uvar infection occurred in both families, but that only the R21 oysters successfully controlled viral replication.

Line 231-233: This MS focuses on hemocytes, what about other potential cells?

Line 271-274: In other parts of the world, POMS also kills adult oysters...

Line 274-276: I don't understand the use of herpesvirus in italics here. Its not just a herpesvirus.

Line 277: please modify as follows "We showed that infection by the OsHV-1 uvar is..." by removing the word "virus". Its already part of the OsHV-1 acronym.

Line 279-280: please modify as follows "We showed that the immune cells of oysters, the hemocytes are a target of the virus, though we did not investigate other cell types."

Line 312-313: It would be wise to indicate that the limitation of the pooled data here—follow-up studies should use look at these genes in individual oysters.

Line 314-316: This figure is only shown in the supplement. Can it be shown in the main body of the MS?

Line 317-319: Please modify as follows "These results..., which are strongly activated in susceptible oysters, may play a role in the success of OshV-1 infection." Must is too strongly worded.

Line 323-326: I believe you should reference previous studies on anti-viral pathways in *C. gigas* exposed to OshV-1. Unless I misunderstand what pathways you are referencing here, other studies have used targeted approaches to antiviral genes in *C. gigas*.

Line 370-372: Please modify as follows "Once oysters...were obtained by gamete stripping..."

Line 386-388: I'm hesitant to call these oysters "pathogen-free" when no histology was conducted and only specific pathogens/pathogen types were targeted. I would remove "Finally, all pathogen-free..." and begin this sentence with "Oysters..."

Line 392-395: what are the size of these oysters?

Line 395: 0.01% seems very low?

Line 397-399: I do not see any evidence of replication here.

Line 524-526: Please provide a citation here.

Line 565-573: What about the qPCR conditions? And the primers used? And the efficiency/R² of the qPCRs?

Reviewer #2 (Remarks to the Author):

This manuscript describes a holistic study of the infectious process, microbiome changes and host responses that are associated with Pacific oyster mortality disease. The authors should be congratulated in taking this integrative step and their data provides some exciting and interesting insights into the multi-infection processes (viral followed by bacterial opportunists). The transcriptome data presented also provides some nice data that speaks to the potential mechanisms by which disease can occur and why some strains maybe more at risk.

Overall this is a well-written manuscript but it would benefit from some minor language and style editing. The discussion could be revised as in parts it tended to repeat or paraphrase much of the results. I would also caution against the use of overstating claims e.g. line 200 - "this massive number".

While I do think the premise of the work is excellent, the main concerns I have with the manuscript is the reporting of the microbiome data. It is important that these data are appropriately analyzed and fully reported on, rather right now it feels more like a surface analysis of the microbiome as an add on to the main aims of study. I have made some specific comments below but I highly recommend that the data be revised by a qualified statistician who is experienced with the analyses of large

multivariate microbiome (and transcriptomic) data sets such as these.

Some specific comments include:

Line 81- re-phrase studied in distinct studies... suggest something like "have been studied with a focus on a restricted number of factors tested in isolation"

Line 87- Please define "full-sib" families when it is first mentioned

Line 86-97 This section could be revised and condensed to be more succinct. The study is valid so I would not necessarily over emphasize the "ecological realistic experimental infection" as it opens up the study to criticism because in the end the experiments are performed under artificial (farmed) conditions. This might rather be a discussion point to highlight that it moves closer to ecologically realistic conditions.

Line 390 and figure S1- While the experimental set-up is clear it is difficult to follow the experimental design with respect to the number of independent replicates and the time-points of collection. I suggest adding these details (number of replicates sampled at each time point for the different families) to Figure S1.

Line 139-173- As mentioned above I have some concerns regarding the accuracy of the statistical tests chosen to analyze the microbiome data. The main issue is what appears to be the use of univariate tests (ANOVA and T-tests) for multivariate datasets with complex designs where some of the data points are not necessarily independent (e.g. time, but see comment above on Fig S1). For example Figure 2 b shows the Bray-Curtis dissimilarity between the replicates of the two oyster groups (R and S) over time in comparison to T0 and is used to conclude that the microbial communities between R and S are statistically different using a T-test for one time only (line 150-159). A more appropriate measure way to display this data could be using a PCoA or MDS ordination so as the readers can visualize the spatial clustering of the data. Then support these observations with the appropriate multivariate statistical method (e.g. PERMANOVA) to assess for differences in B-diversity. A measure of dispersion (PERMDISP) could then be used to determine if there was more variation (dispersion) in the communities between replicates in one of the treatments than the other.

Line 258 Describes a separate microbiome study of diseased individuals resulting from experiments with either vibrio or viral infection (Figure 7). These results are depicted in Figure S11 as a series of selected OTU abundances. It is unclear exactly how these OTU's were chosen and the analysis itself appears incomplete (at the very least the reporting of it is incomplete). For example could the authors detect *V. crassostreae* sequences in diseased conditions? I would expect it to be present at least in the inoculated individuals. Also an important control that is missing is the microbiome of healthy individuals, did these differ significantly in their microbial community structure? or were the difference only in the few highly abundant (>4%) taxa indicated?

This is important as the data is used to support the claim that bacteria other than *Vibrio*'s are capable of causing secondary bacterial infections (line 301- 307).

Line 180 and Figure S7. It is unclear if the confirmatory qPCR experiments were performed on the same biological material as the RNA-Seq data or if these were independent replicate experiments.

Line 405- please indicate the frequency of the sampling

Figure 1. line 846 change "plain line" to "solid line". There is no indication of the number of replicates

or variability of resistance given in these figures. It is also very difficult to make out the values in figure 1b for the resistant phenotype. Maybe consider absolute values rather than % survival and indicate in the legend number of replicates or number of times the experiment was reproduced and in the figure the variability. Without this information Figure 1B also becomes redundant to what is already in the text (e.g. line 119)

Line 575-603- Please indicate in the methods and relevant figures to what % identity the OTU's were clustered.

Line 595- 597, can you further clarify what is meant by this sentence and what exactly was removed?

Line 598 talks using relative abundances of taxa to account for variation in sequencing depth. However the more appropriate way to do this might be subsample/ rarefy the data to the lowest number of counts. Alternatively count data could be used as a co-variable when using Generalized Linear Modeling to statistically analyze the data for B- diversity (but not Alpha-diversity measure where rarefied data is needed). It is not clear from the methods what exactly has been done.

Line 624 mentions alpha diversity measurements but I did not see these reported in the manuscript.

Line 423- Please re-phrase English here to "co-habitation experiment was performed by exposing" Also indicate the number of replicates for each experiment control and treatment.

Line 428- refers to Supplementary data Figure S12. It is not clear why results are being described in the methods section. Also Figure S12 does not include any information on the variation of the experiment are the values in the bar graphs averages of % survival from a number of experimental replicates? The legend of Figure S12 also refers of Figure 7, but again there is no indication from Figure a, f and k of the level of variation in mortality.

Line 430- Please clarify the design for these experiments. What is meant by "decoupled experimental experiments"?

Throughout the paper please refer to 16S rRNA gene (or 16S rDNA) rather than 16S (e.g. but not limited to lines 497, 519)

Please make the color key associated with each of the heatmap figures clearer and explain what this is in the legend. What is the histogram? This was almost impossible to see.

Responses to reviewers' comments:

Reviewer #1:

Pacific oyster mortality syndrome is a disease of major significance to Pacific oyster culture globally. The authors use a multi-pronged approach to study the disease in resistant and susceptible seed Pacific oyster families to study the course of POMS. The major claim of the MS is that OsHV-1 is the primary causative agent but secondary bacterial infections are an important contributing factor in mortalities. One of the biggest strengths is the combination of experimental biology with both classic (histology) and modern tools (i.e. next generation sequencing, 16S tagged sequencing, ISH) focused on both the host and pathogens. I believe this MS will influence other researchers in the field, but has major limitations,

1-1 i.e. the lack of replication in experimental approaches and pooling of animals for molecular analyses. I appreciate the use of both 'natural' and 'rationalized' experimental approaches, though both approaches lack replication, which concerns me about repeatability.

To address this concern, we have included replicates from real independent experiments in the new version of the manuscript.

- Concerning the "natural" experimental approach, we have included data from two other resistant (R_{F23} and R_{F48}) and two other susceptible (S_{F14} and S_{F15}) families that were exposed to the same natural infectious environment. As in the first dataset, 3 pools of ten individuals were used for each family. Taken together our data validate the early intense replication of the virus and the subsequent bacteremia in the three distinct susceptible families of oysters. We also confirmed this sequence of events did not occur in three distinct families of resistant oysters. These data are shown in a new Figure (Fig. 9a, b, c & d). In addition, we also measured gene expression of AMPs and *Cg*-SOD and IAPs (from the host and the virus) for the 4 additional families (Fig. 9e & 9f). We also showed expression data for some genes representative of the antiviral response on these different families and confirmed the early and efficient antiviral response that is a common feature of all the resistant families (Fig. 9g). All these results are included in a new paragraph of the result section; this paragraph is entitled "Similar molecular events occurred in other susceptible and resistant oyster families". These additional experiments confirm the sequence of events of the pathogenesis in susceptible oysters and the mechanisms underlying resistance that were presented only for two families in the previous version of the manuscript.

- Concerning the "rationalized" experimental approach, two additional independent experiments were performed using oysters from a distinct genetic background (another biparental family of susceptible oysters). Virus/bacteria were monitored using the same protocol presented in the first version of the manuscript. The same results were obtained in these two additional experiments: 1) PolyI:C blocked the viral replication and the subsequent bacteremia leading to oyster death and 2) the antibiotic treatment significantly delayed and reduced mortalities. This confirms the scheme presented in the first version of the manuscript. Data are included in the Supplementary Figure 13 of the revised manuscript.

The lack of replication was also pointed by the second reviewer and we hope that these new data in our revised version of the manuscript is satisfactory for both reviewers.

1-2 I am also concerned with pooling animals for molecular analyses. If the oysters are large enough to inject, achieving high RNA quality from individuals should not be an issue; in fact pooling may include individuals that are not of high enough quality.

The “natural” experiments were performed on quite small animals. The mean individual weight was ~1.10 g which corresponds to a flesh weight without shell of ~0.2 g (the oysters weight was added in the methods section). Samples were subsequently treated for DNA and RNA extraction, as needed for 16S metabarcoding, qPCR monitoring of pathogens, RT-qPCR monitoring of oyster genes and RNAseq. Such an exhaustive monitoring of the oyster holobiont required enough material preventing us from doing individual analyses. Moreover, our study relies on a substantial number of replicates, kinetic times and molecular analyses. For cost reasons, individual analysis would have prevented us from making such a high number of analyses.

In addition, the infections were performed on biparental families and the ten individuals from each family share homogenous genotypes and phenotypes (resistant or susceptible). As a consequence, pooling ten individuals should not influence the result as it would be the case when the animals are genetically diversified.

For all these reasons, we decided to use pools of animals.

The new results included in the manuscript (on two additional resistant families and two additional susceptible families) demonstrate the robustness of the pathogenesis scheme, and compensate for our experimental choice (pools instead of individuals).

1-3 The statistical analyses appear complete and appropriate for the data, except further data might be interesting to show within the supplementary files, in particular the fold change of genes of interest for the oyster—I'm curious to the underlying fold change differences (beyond the enrichment analysis) and also a heatmap of the viral gene expression over time.

The requested fold changes of genes of interest (beyond the enrichment analysis) are shown in Supplementary Table 4. As requested, we included in the revised manuscript a heatmap of the viral gene expression over the “natural” experimental infection (Supplementary Fig. 2). A sentence introducing Supplementary Fig. 2 was added in the corresponding paragraph of the result section of the revised manuscript.

1-4 The authors use two experiment types; it would be helpful to the reader to define the experiment types as separate headings in the results and/or on the figures.

To clarify the separation between “natural” and rationalized experiments in the text and the figures of the revised manuscript, we added the terms “rationalized experimental infections” in the title of the Figure 10, Supplementary Figure 12, 13, 14 and 15. The term “rationalized experimental infections” was also added in the title of the corresponding paragraph of the result section (last paragraph). The term “natural” experimental infection is used all over the manuscript (text and figure) for the first experiment type.

Specific line by line comments:

1-5 Line 71-74: The authors need to take care in how they describe OsHV-1 uvar. We don't currently have a good understanding of the number of microvariant strains globally; additionally the OIE only defines a specific type strain as OsHV-1 uvar.

The sentence “an *Ostreid herpesvirus* variant (called OsHV-1 μ Var)” was replaced by “*Ostreid herpesvirus* variants” and 3 additional recent references describing the diversity of *Ostreid herpesvirus* were added in the text.

1-6 Line 93-97: I suggest defining POMS much earlier in the introduction and provide a citation for the use of this acronym.

As recommended, this term was defined in the introduction section and the following references was added: Paul-Pont, I., Dhand, N. K. & Whittington, R. J. Influence of husbandry practices on OshV-1 associated mortality of Pacific oysters *Crassostrea gigas*. *Aquaculture* **412**, 202-214, (2013).

1-7 Line 133-135: How did you define this is an OshV-1 uvar?? Please include the specific modifications of the genome that defined this variant as OshV-1 uvar. A table in the supplement describing the modifications would be helpful.

For more clarity, the following sentences describing the specific modifications in the genome were added in the appropriate paragraph of the result section: "Alignment of the Illumina reads to the OshV-1 genome available in NCBI database³⁵ revealed that the virus involved in our experiments corresponds to an OshV-1 μ Var variant as indicated by (i) the deletion of 3 ORFs (ORF36, ORF37 and part of ORF38), (ii) the deletion of an adenosine upstream of ORF43 and (iii) a 12 nt deletion in the microsatellite locus H10, which are shared characteristics of μ Var variants (Segarra *et al.*, 2010, Renault *et al.*, 2014, Burioli *et al.*, 2017). "

1-8 Line 135-137: I would suggest the following modification "Taken together, these results indicate a successful OshV-1 uvar infection occurred in both families, but that only the R21 oysters successfully controlled viral replication.

The proposed modification was done.

1-9 Line 231-233: This MS focuses on hemocytes, what about other potential cells?

Previous studies investigated the global distribution of the virus in tissues of infected oysters (Martenot *et al.*, *J Invertebrate Pathol*, 136 (2016)124-135). In the present work, the objective of *in situ* hybridization experiments was to verify that the main cells involved in the antibacterial defense, the hemocytes, were infected by the virus by showing the colocalization of OshV-1 and SOD (specific marker of hemocytes) labelling. We observed that cells other than hemocytes were also infected but as this result was not useful for the discussion, photographs were not displayed here.

1-10 Line 271-274: In other parts of the world, POMS also kills adult oysters...

To address this comment, the following sentence "In this study, we deciphered the complex intra-host interactions underlying the mortality syndrome affecting juveniles of *Crassostrea gigas*, providing a comprehensive view of the pathogenic processes underpinning a disease that has remained incompletely understood until now." was replaced by "In this study, we deciphered the complex intra-host interactions underlying the POMS, providing a comprehensive view of the pathogenic processes underpinning a disease that has remained incompletely understood until now."

1-11 Line 274-276: I don't understand the use of herpesvirus in italics here. Its not just a herpesvirus.

Italics were removed.

1-12 Line 277: please modify as follows "We showed that infection by the OshV-1 uvar is..." by removing the word "virus". Its already part of the OshV-1 acronym.

The proposed modification was done.

1-13 Line 279-280: please modify as follows "We showed that the immune cells of oysters, the hemocytes are a target of the virus, though we did not investigate other cell types.

The following modification was added to the text: "We showed that the immune cells of oysters, the hemocytes, are one of the cell types targeted by the virus."

1-14 Line 312-313: It would be wise to indicate that the limitation of the pooled data here—follow-up studies should use look at these genes in individual oysters.

We answered the criticism above in the 1-2 section of this response to reviewer 1.

1-15 Line 314-316: This figure is only shown in the supplement. Can it be shown in the main body of the MS?

As requested, this figure was included in the main body of the manuscript (Figure 6). To improve the visualization, histograms were transformed into heatmaps.

1-16 Line 317-319: Please modify as follows "These results..., which are strongly activated in susceptible oysters, may play a role in the success of OsHV-1 infection." Must is too strongly worded.

The proposed modification was done.

1-17 Line 323-326: I believe you should reference previous studies on anti-viral pathways in *C. gigas* exposed to OsHV-1. Unless I misunderstand what pathways you are referencing here, other studies have used targeted approaches to antiviral genes in *C. gigas*.

We agree that antiviral signalling pathways were shown to be induced in oysters in previous works: some of these references were given in the introduction section of this manuscript (He Y. et al 2015, Segarra A. et al 2014, Rosani et al 2017). To make that point clear, the reference of a review (Green TJ et al 2015) was added in the introduction of the revised manuscript. The novel and original result from our study is the early induction of these pathways in resistant oyster families (by comparison with the susceptible families in which the induction is delayed).

1-18 Line 370-372: Please modify as follows "Once oysters...were obtained by gamete stripping..."

The proposed modification was done.

1-19 Line 386-388: I'm hesitant to call these oysters "pathogen-free" when no histology was conducted and only specific pathogens/pathogen types were targeted. I would remove "Finally, all pathogen-free..." and begin this sentence with "Oysters..."

The proposed modification was done and when pathogen-free was used inverted comas were included ("pathogen-free") all along the manuscript.

1-20 Line 392-395: what are the size of these oysters?

As requested, the mean individual flesh weight (0.30 g) was added in the text.

1-21 Line 395: 0.01% seems very low?

This corresponds to 6 dead oysters on the total number of donors deployed in the farming area (57.000). As no mortality were observed from the beginning of the exposure until these first mortalities, this signal was sufficient to be certain that the oysters were diseased. This procedure was applied successfully in several studies (Petton et al 2013, Petton et al 2015, Le Roux et al. 2016, Bruto et al. 2017).

1-22 Line 397-399: I do not see any evidence of replication here.

The criticism about replication was addressed above. We have included the analysis of 4 additional families (2 resistant and 2 susceptible ones) in the present revised work.

1-23 Line 524-526: Please provide a citation here.

The following reference was added in the text: Gonzalez, M. *et al.* Evidence of a bactericidal permeability increasing protein in an invertebrate, the *Crassostrea gigas* Cg-BPI. *Proc Natl Acad Sci U S A* 104, 17759-17764, (2007).

1-24 Line 565-573: What about the qPCR conditions? And the primers used? And the efficiency/R² of the qPCRs?

All these data were included in the revised manuscript. qPCR conditions were added in the methods section. The primers and PCR efficiencies are given in the Supplementary Table 8 and Supplementary Table 9.

Reviewer #2:

This manuscript describes a holistic study of the infectious process, microbiome changes and host responses that are associated with Pacific oyster mortality disease. The authors should be congratulated in taking this integrative step and their data provides some exciting and interesting insights into the multi-infection processes (viral followed by bacterial opportunists). The transcriptome data presented also provides some nice data that speaks to the potential mechanisms by which disease can occur and why some strains maybe more at risk.

2-1 Overall this is a well-written manuscript but it would benefit from some minor language and style editing.

All language and style modifications proposed by the reviewers were done. In addition, the manuscript was edited by the Nature Research English Language Editing Service (<http://bit.ly/NRES-LS>).

2-2 The discussion could be revised as in parts it tended to repeat or paraphrase much of the results.

As proposed, the discussion section was revised and shortened in order to avoid redundancy with the results section.

2-3 I would also caution against the use of overstating claims e.g. line 200 - "this massive number".

"a massive number" was replaced by "a large number".

2-4 While I do think the premise of the work is excellent, the main concerns I have with the manuscript is the reporting of the microbiome data. It is important that these data are appropriately analyzed and fully reported on, rather right now it feels more like a surface analysis of the microbiome as an add on to the main aims of study. I have made some specific comments below but I highly recommend that the data be revised by a qualified statistician who is experienced with the analyses of large multivariate microbiome (and transcriptomic) data sets such as these.

We have improved the microbiome analyses by addressing all the specific comments raised by the reviewer (see details below). Analyses of microbiome dataset were supervised by Dr Camille Clerissi (author of this manuscript). He is a qualified statistician experienced in the analyses of large NGS data sets; he published papers based on multivariate analyses in high quality journals (Clerissi et al. 2014. AEM, Clerissi et al. 2014. Virology, Clerissi et al. 2015. Env Microbiol Reports, Brener-Raffalli et al. 2018. Microbiome, Clerissi et al., Nature Communications 2018). We explain below (please see comment 2.9) our previous experimental and analytical choices and how we revised the analyses as requested. Note that our main conclusions did not change between the previous and the revised version of the manuscript. In addition, to support the conclusions, we added to the manuscript an analysis of alpha diversity as classically presented in microbiota studies.

Some specific comments include:

2-5 Line 81- re-phrase studied in distinct studies... suggest something like" have been studied with a focus on a restricted number of factors tested in isolation"

The sentence was modified as proposed by the reviewer.

2-6 Line 87- Please define "full-sib" families when it is first mentioned

The term "full-sib family" was replaced by "biparental families" all along the manuscript.

2-7 Line 86-97 This section could be revised and condensed to be more succinct. The study is valid so I would not necessarily over emphasize the "ecological realistic experimental infection" as it opens up the study to criticism because in the end the experiments are performed under artificial (farmed) conditions. This might rather be a discussion point to highlight that it moves closer to ecologically realistic conditions.

As requested the term "ecological realistic experimental infection" was removed in this paragraph and in all sections of the manuscript. It was replaced by "natural experimental infections", as proposed by the first reviewer. Inverted commas ("natural") were used all along the manuscript to avoid the present criticism. As requested, the paragraph (line 86-97) was shortened in the revised manuscript.

2-8 Line 390 and figure S1- While the experimental set-up is clear it is difficult to follow the experimental design with respect to the number of independent replicates and the time-points of collection. I suggest adding these details (number of replicates sampled at each time point for the different families) to Figure S1.

In order to clarify this point, the following sentence was added to the text in the corresponding paragraph of the methods section: "During the experimental infection, 3 triplicates of 10 oysters were sampled from each tank and each time (0, 6 h, 12 h, 24h,

48 h, 60 h and 72h) of the kinetics". As requested, the same sentence was added to the legend of the Figure S1.

2-9 Line 139-173- As mentioned above I have some concerns regarding the accuracy of the statistical tests chosen to analyze the microbiome data. The main issue is what appears to be the use of univariate tests (ANOVA and T-tests) for multivariate datasets with complex designs where some of the data points are not necessarily independent (e.g. time, but see comment above on Fig S1). For example Figure 2 b shows the Bray-Curtis dissimilarity between the replicates of the two oyster groups (R and S) over time in comparison to T0 and is used to conclude that the microbial communities between R and S are statistically different using a T-test for one time only (line 150-159). A more appropriate measure way to display this data could be using a PCoA or MDS ordination so as the readers can visualize the spatial clustering of the data. Then support these observations with the appropriate multivariate statistical method (e.g. PERMANOVA) to assess for differences in B-diversity. A measure of dispersion (PERMDISP) could then be used to determine if there was more variation (dispersion) in the communities between replicates in one of the treatments than the other.

Analyses were redone according to the recommendation of referee #2. In particular, we based normalization and microbial differential abundance strategies on the recent paper published by Weiss et al. (2017, *Microbiome*, doi:10.1186/s40168-017-0237-y). Hence, we first rarefied microbiome dataset. Secondly, we computed a PCoA to represent the spatial clustering of data instead of using a kinetic plot. Thirdly, we used a measure of dispersion to estimate differences in term of dispersions between the two oyster families. Fourthly, we used DESeq2 to identify taxa that had different abundances between initial and final time points of the kinetic. Fifthly, we added alpha diversity index and their corresponding statistical tests.

2-10 Line 258 Describes a separate microbiome study of diseased individuals resulting from experiments with either vibrio or viral infection (Figure 7). These results are depicted in Figure S11 as a series of selected OTU abundances. It is unclear exactly how these OTU's were chosen and the analysis itself appears incomplete (at the very least the reporting of it is incomplete). For example could the authors detect *V. crassostreae* sequences in diseased conditions? I would expect it to be present at least in the inoculated individuals.

Analyses of significant changes in bacterial community composition between T0 and T72h were performed at the genus level because many OTUs could not be affiliated to the species level using 16S rDNA metabarcoding. Thus, this methodology did not allow testing for changes in *V. crassostreae* proportions. Instead, this species was monitored using quantitative PCR and detected in *V. crassostreae*-inoculated samples only (Figure 10).

2-11 Also an important control that is missing is the microbiome of healthy individuals, did these differ significantly in their microbial community structure? or were the difference only in the few highly abundant (>4%) taxa indicated?

This is important as the data is used to support the claim that bacteria other than *Vibrio*'s are capable of causing secondary bacterial infections (line 301- 307).

As requested, we added the microbiome of healthy individuals as a control. Only one genus significantly differed between T0 and T72h (Supplementary Figure 15f).

In the first version of the manuscript, we showed highly abundant taxa (>4%) whose abundance varied significantly. In order to address this criticism, we added to the manuscript all taxa that differed significantly between T0 and T72h without applying any abundance cut-off (Supplementary Table 7). As the same criticism can be done for the

Figure 2b (“natural” infection experiment), all taxa that were significantly modified were also shown in a new Supplementary Table (3).

2-12 Line 180 and Figure S7. It is unclear if the confirmatory qPCR experiments were performed on the same biological material as the RNA-Seq data or if these were independent replicate experiments.

As a technical validation of our RNAseq, the RT-qPCR analysis was indeed performed on the same RNAs than those used for RNAseq analysis. For more clarity, we have modified the following sentence in the figure legend (Supplementary Figure 9): “their relative expressions were quantified by RT-qPCR on the same for the RNA-seq approach.”

2-13 Line 405- please indicate the frequency of the sampling

The sampling times were added to the text and to the legend of figure 1 to clarify this point.

2-14 Figure 1. line 846 change “plain line” to “solid line”. There is no indication of the number of replicates or variability of resistance given in these figures. It is also very difficult to make out the values in figure 1b for the resistant phenotype. Maybe consider absolute values rather than % survival and indicate in the legend number of replicates or number of times the experiment was reproduced and in the figure the variability. Without this information Figure 1B also becomes redundant to what is already in the text (e.g. line 119).

As proposed, the figure 1 was modified (1b was removed). The legend was also completed as follows: “Production of oyster biparental families with contrasted resistance phenotypes. Kaplan-Meier survival curves of the 15 families of recipient oysters during the “natural” experimental infection. At each time indicated on the arrow, 3 triplicates of 10 oysters were sampled from each tank for further molecular analysis”.

2-15 Line 575-603- Please indicate in the methods and relevant figures to what % identity the OTU's were clustered.

As indicated in the methods section, we performed OTU clustering with the SWARM algorithm, which uses a local clustering threshold (aggregation distance) instead of a global similarity threshold. The reference Mahe et al. 2014 that describes the clustering was added to the revised version of the manuscript.

2-16 Line 595- 597, can you further clarify what is meant by this sentence and what exactly was removed?

The following sentence was added at the end of the corresponding methods section: “In our analyses, we only kept taxa having a true annotation for each corresponding taxonomic rank (from genus to phylum).”

2-17 Line 598 talks using relative abundances of taxa to account for variation in sequencing depth. However the more appropriate way to do this might be subsample/ rarefy the data to the lowest number of counts. Alternatively count data could be used as a co-variable when using Generalized Linear Modeling to statistically analyze the data for B- diversity (but not Alpha-diversity measure where rarefied data is needed). It is not clear from the methods what exactly has been done.

Answers to these comments are given in the answer to 2.9 comment. Briefly, in the revised manuscript, we based normalization and microbial differential abundance strategies on the recent paper published by Weiss et al. (2017, Microbiome, doi:10.1186/s40168-017-0237-y). We first rarefied microbiome dataset. Secondly, we computed a PCoA to represent the spatial clustering of data instead of using a kinetic plot. Thirdly, we used a measure of dispersion to estimate differences in term of dispersions between the two oyster families. Fourthly, we used DESeq2 to identify taxa that had different abundances between initial and final time points of the kinetic.

2-18 Line 624 mentions alpha diversity measurements but I did not see these reported in the manuscript.

Indeed, analyses of alpha diversity indices were not displayed in the previous version of the manuscript. In the revised manuscript, we added Chao1 and Shannon indexes, and we computed univariate statistical tests to describe their modifications along the kinetic (Supplementary Figure 6).

2-19 Line 423- Please re-phrase English here to "co-habitation experiment was performed by exposing" Also indicate the number of replicates for each experiment control and treatment.

As mentioned above, the revised manuscript has been edited by the Nature Research English Language Editing Service.

Moreover, as indicated in the response to reviewer 1 comment (comment 1-1), we performed two additional independent rationalized experimental infections (Supplementary Fig 13). The following sentence was added at the end of the paragraph: "Two additional rationalized experimental infections were performed using the same design on susceptible juveniles from a biparental family (family H12¹⁰) (Supplementary Fig. 13). Both experiments confirmed the essential role of OsHV-1 replication in bacterial colonization and death of recipient oysters (mortality data of donors are given in Supplementary Fig. 14)."

2-20 Line 428- refers to Supplementary data Figure S12. It is not clear why results are being described in the methods section. Also Figure S12 does not include any information on the variation of the experiment are the values in the bar graphs averages of % survival from a number of experimental replicates? The legend of Figure S12 also refers of Figure 7, but again there is no indication from Figure a, f and k of the level of variation in mortality.

As requested, these results were included in the last paragraph of the result section (entitled "Rationalized experimental infections revealed that OsHV-1 replication is necessary for bacterial colonization leading to oyster death."). The mortalities of donors were followed all along the experiment. End-point mortalities (72 h) are displayed in Supplementary Figure 12 for the first experiment and Supplementary Figure 14 for the 2 additional experiments .

2-21 Line 430- Please clarify the design for these experiments. What is meant by "decoupled experimental experiments"?

"Decoupled experimental experiments" was replaced by "rationalized experimental infections" in the revised manuscript.

2-22 Throughout the paper please refer to 16S rRNA gene (or 16S rDNA) rather than 16S (e.g. but not limited to lines 497, 519)

The modification was done all along the manuscript.

2-23 Please make the color key associated with each of the heatmap figures clearer and explain what this is in the legend. What is the histogram? This was almost impossible to see.

The color key was modified. The intensity of the blue color represents the frequency of taxa. This has been indicated in the figure legends.

Certificate of Nature Research English Language Editing Service for revision of English

SPRINGER NATURE | Author Services

Nature Research Editing Service Certification

This is to certify that the manuscript titled *Cracking the code of Pacific oyster mortality syndrome* was edited for English language usage, grammar, spelling and punctuation by one or more native English-speaking editors at Nature Research Editing Service. The editors focused on correcting improper language and rephrasing awkward sentences, using their scientific training to point out passages that were confusing or vague. Every effort has been made to ensure that neither the research content nor the authors' intentions were altered in any way during the editing process.

Documents receiving this certification should be English-ready for publication; however, please note that the author has the ability to accept or reject our suggestions and changes. To verify the final edited version, please visit our verification page. If you have any questions or concerns over this edited document, please contact Nature Research Editing Service at support@as.springernature.com.

Manuscript title: Cracking the code of Pacific oyster mortality syndrome

Authors: de Lorgeril et al.

Key: 1CA1-C747-15AB-50E7-D9AP

This certificate may be verified at secure.authorservices.springernature.com/certificate/verify.

Nature Research Editing Service is a service from Springer Nature, one of the world's leading research, educational and professional publishers. We have been a reliable provider of high-quality editing since 2006.

Nature Research Editing Service comprises a network of more than 900 language editors with a range of academic backgrounds. All our language editors are native English speakers and must meet strict selection criteria. We require that each editor has completed or is completing a Masters, Ph.D. or M.D. qualification, is affiliated with a top US university or research institute, and has undergone substantial editing training. To ensure we can meet the needs of researchers in a broad range of fields, we continually recruit editors to represent growing and new disciplines.

Uploaded manuscripts are reviewed by an editor with a relevant academic background. Our senior editors also quality-assess each edited manuscript before it is returned to the author to ensure that our high standards are maintained.

CERTIFICATE 1CA1-C747-15AB-50E7-D9AP - JUNE 21, 2018 - PAGE 1 OF 1

REVIEWERS' COMMENTS:

Reviewer #1 (Remarks to the Author):

**In my previous review, I provided the following summary
"Pacific oyster mortality syndrome is a disease of major significance to Pacific oyster culture globally. The authors use a multi-pronged approach to study the disease in resistant and susceptible seed Pacific oyster families to study the course of POMS. The major claim of the MS is that OsHV-1 is the primary causative agent but secondary bacterial infections are an important contributing factor in mortalities. One of the biggest strengths is the combination of experimental biology with both classic (histology) and modern tools (i.e. next generation sequencing, 16S tagged sequencing, ISH) focused on both the host and pathogens. I believe this MS will influence other researchers in the field, but has major limitations"**

In the previous review I expressed several limitations (labeled as 1-1 to 1-4 in the review response). I am satisfied with the responses I received, and am impressed with the thoroughness of the authors response to reviews. Though the main premise of the MS did not change, I believe it to be stronger now and will have a greater impact on the field.

Here are minor comments to consider.

Abstract

Line 49: Why say "herpesvirus". Why not give its name.

Line 67-69: "Exploited" has a certain connotation of negativity, and I don't believe this is the intent. I would suggest changing "exploited" to "harvested" or "utilized".

Line 447-449: How were the oysters chosen to sample, since they were "sampled without randomization and blinding protocols from each tank and each time..." ? (I can only imagine that to truly randomize and blindly chose oysters, they would have to be housed in numbered individual compartments within each tank). Was there a selection criteria? Live versus dead or just simply "randomly" selecting oysters?

Reviewer #2 (Remarks to the Author):

I thank the authors for the time they have taken to improve their manuscript according to my initial suggestions. This study is really interesting, I am happy with the change they have made and have no further comments to make.

RESPONSES TO REVIEWERS' COMMENT

Reviewer #1:

In my previous review, I provided the following summary "Pacific oyster mortality syndrome is a disease of major significance to Pacific oyster culture globally. The authors use a multi-pronged approach to study the disease in resistant and susceptible seed Pacific oyster families to study the course of POMS. The major claim of the MS is that OsHV-1 is the primary causative agent but secondary bacterial infections are an important contributing factor in mortalities. One of the biggest strengths is the combination of experimental biology with both classic (histology) and modern tools (i.e. next generation sequencing, 16S tagged sequencing, ISH) focused on both the host and pathogens. I believe this MS will influence other researchers in the field, but has major limitations". In the previous review I expressed several limitations (labeled as 1-1 to 1-4 in the review response). I am satisfied with the responses I received, and am impressed with the thoroughness of the authors response to reviews. Though the main premise of the MS did not change, I believe it to be stronger now and will have a greater impact on the field.

Here are minor comments to consider.

Line 49: Why say "herpesvirus". Why not give its name.

As suggested, the name of the virus is now used in the text.

Line 67-69: "Exploited" has a certain connotation of negativity, and I don't believe this is the intent. I would suggest changing "exploited" to "harvested" or "utilized".

As suggested, « utilized » is used instead of « exploited ».

Line 447-449: How were the oysters chosen to sample, since they were "sampled without randomization and blinding protocols from each tank and each time..." ? (I can only imagine that to truly randomize and blindly chose oysters, they would have to be housed in numbered individual compartments within each tank). Was there a selection criteria? Live versus dead or just simply "randomly" selecting oysters?

Oysters were just « randomly selected ». At this stage of the experiment, the sampled oysters were all alive. The text was modified as follow: "10 oysters in triplicate were randomly sampled without blinding protocols".

Reviewer #2

I thank the authors for the time they have taken to improve their manuscript according to my initial suggestions. This study is really interesting, I am happy with the change they have made and have no further comments to make.